# MultiHaystack: Benchmarking Multimodal Reasoning over 40K Images, Videos, and Documents

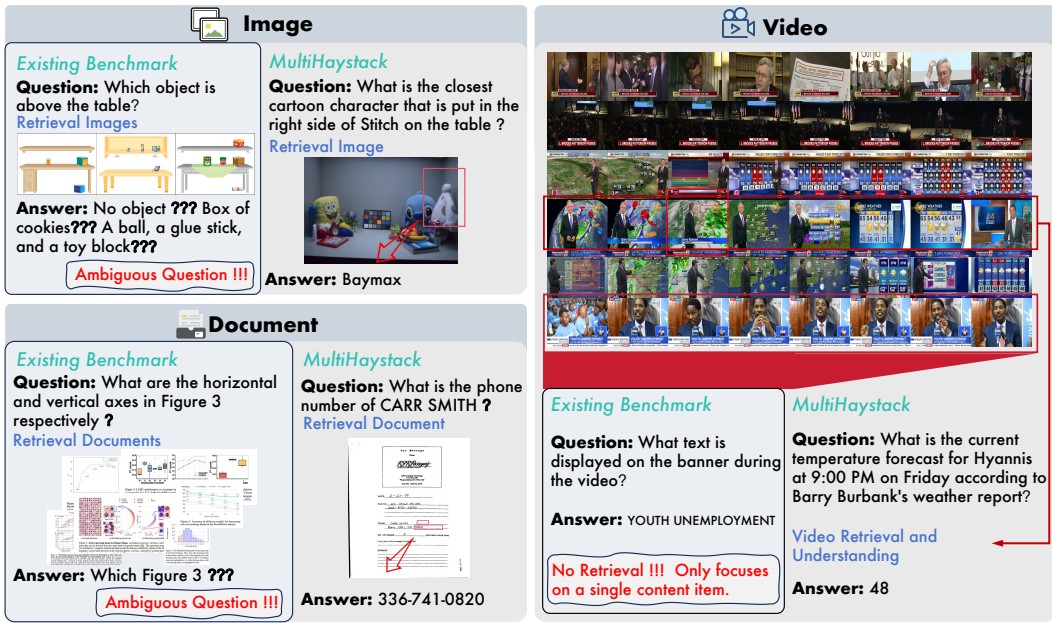

Figure 1: **Comparison of visual question answering benchmarks.** Existing benchmarks focus on isolated content (e.g., a single image or video), lack large-scale retrieval, and often pose ambiguous questions. In contrast, **MultiHaystack** provides retrieval-oriented, evidence-grounded queries across documents, images, and videos, requiring modality selection and fine-grained reasoning to better reflect real-world scenarios.

## Abstract

Multimodal large language models (MLLMs) have advanced rapidly on benchmarks involving isolated text, image, or video tasks, but such settings overlook a crucial step in real-world applications: retrieving evidence from large, heterogeneous corpora before reasoning. Existing benchmarks typically provide only hundreds or thousands of candidates, making retrieval trivial and overstating model reliability. To address this gap, we introduce MultiHaystack, the first benchmark for large-scale, realistic cross-modal retrieval and reasoning. It contains over 46,000 documents, images, and videos paired with 747 uniquely verifiable questions, ensuring unambiguous evaluation while requiring both modality selection and fine-grained reasoning. Our experiments reveal a consistent pattern: models perform competitively when directly given the answer-containing file, but their performance drops sharply once evidence must be retrieved at scale. The best retriever (E5-V) achieves only 40.8% Recall@1, while even GPT-5 reaches just 51.4% VQA accuracy under top-5 retrieval. These results reveal that retrieval, rather than reasoning, is the dominant bottleneck, establishing MultiHaystack as a rigorous benchmark that exposes weaknesses hidden by small-scale evaluations and highlights retrieval as the key frontier for advancing MLLMs.

# 1 INTRODUCTION

Multimodal large language models (MLLMs) have advanced rapidly by unifying text, image, and video within a single generative framework (Chen et al., 2023). They support applications ranging from visual assistants (Liu et al., 2025; Zhu et al., 2025) to Retrieval-Augmented Generation (RAG) (Yu et al., 2025). However, most evaluations focus only on reasoning, overlooking the critical step that precedes it in practice: retrieving relevant evidence from large heterogeneous corpora spanning documents, images, and videos. This gap raises the central question: *when large-scale cross-modal retrieval is required, can current MLLMs still retrieve and reason reliably?*

Real-world applications highlight this challenge: users interact with large heterogeneous corpora (*e.g.*, thousands of contracts, hundreds of diagrams, or hours of instructional video) and require retrieving unique evidence and verifying cross-modal consistency. For instance: "What is the penalty clause on page 7, and does it align with step 3 in the video?" Answering such queries requires first retrieving the correct page and video segment from a large candidate pool, then reasoning jointly over them. In contrast, most existing benchmarks focus on isolated content (e.g., a single image or video), which can lead to vague questions when answering requires multimodal evidence (see Figure 1), thereby overstating model reliability. Practical scenarios demand large-scale multimodal retrieval tightly coupled with rigorous evidence-grounded reasoning.

Recent retrieval-oriented datasets have made progress but remain insufficient in three critical aspects. *(1) Unrealistic scale:* most provide only hundreds or thousands of candidates, making retrieval trivial and inflating accuracy (Meng et al., 2025). *(2) Limited modality coverage:* restricted to a single modality and do not adequately assess cross-modal performance, especially the more demanding task of retrieving and integrating evidence across text, images, and videos for reasoning (Chen et al., 2024; Wang et al., 2025). *(3) Ambiguous supervision:* others permit vague or non-unique questions, hindering reproducibility and masking true weaknesses (Mathew et al., 2021). Together, these gaps prevent faithful assessment of MLLM reliability under large-scale, cross-modal retrieval, motivating a benchmark with multimodal corpora and uniquely verifiable questions.

To address these gaps, we introduce MultiHaystack, the first large-scale benchmark for realistic cross-modal retrieval and reasoning. MultiHaystack spans 46,260 documents, images, and videos, paired with 747 evidence-grounded questions. Each question requires retrieving a single relevant item from a multimodal pool of up to 46K candidates, followed by fine-grained reasoning across modalities. Notably, every query is evidence-grounded with a uniquely verifiable answer, ensuring clarity and reproducibility. Our experiments reveal a consistent pattern: accuracy is high when models are given the answer-containing file, but drops sharply once large-scale retrieval is required, especially across modalities. Even strong retrievers such as E5-V achieve only 40.8% Recall@1, while GPT-5 reaches 51.4% VQA accuracy under top-5 retrieval. We further observe a steep performance collapse as the candidate pool expands from 1K to 46K, a challenge that prior small-scale benchmarks fail to expose. These results indicate that *retrieval*, rather than reasoning, is the dominant bottleneck, explaining why small-scale evaluations have masked this fundamental problem (Figure 2).

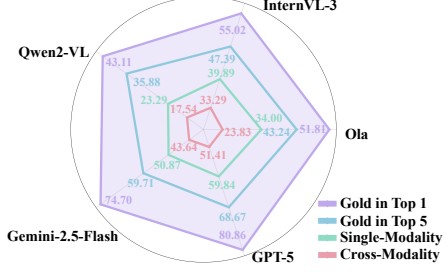

Figure 2: **Performance on MultiHaystack.** "Gold in Top-1/5" directly provides answer-containing files; "Single-Modality" and "Cross-Modality" require retrieval within one or across multiple modalities.

In summary, our contributions are as follows:

- We introduce **MultiHaystack**, the first large-scale benchmark for realistic cross-modal retrieval and reasoning, spanning 46K documents, images, and videos.
- We design evidence-grounded questions that require retrieval with modality selection and cross-modal fusion, enabling precise and reproducible evaluation.
- We conduct comprehensive experiments on state-of-the-art MLLMs, highlighting retrieval as the key frontier for advancing multimodal reasoning.

Table 1: **Comparison of Needle-in-a-Haystack Benchmarks.** Unlike prior benchmarks that are limited in modality, scale, or task diversity, MultiHaystack is a large-scale multimodal benchmark with over 46k contexts, multi-needle retrieval, unique answers, and six task types.

| Benchmark | Modality | Images per QA | Data Types | Multi-Needle | Unique-Answer | Task Types |
|---|---|---|---|---|---|---|
| WebVQA (Doe & Smith, 2023) | | 1–5 | Web images | ✘ | ✔ | 1 |
| RetVQA (Penamakuri et al., 2023) | | 20–30 | Natural images | ✔ | ✔ | 2 |
| MM-NIAH (Wang et al., 2024b) | | 10–70+ | Mixed text-image | ✔ | ✘ | 3 |
| MMNeedle (Wang et al., 2025) | | 10–160 | Image patch | ✔ | ✔ | 1 |
| DocHaystack (Chen et al., 2024) | | 100–1000 | Document images | ✘ | ✔ | 2 |
| **MultiHaystack** | | **>46,000** | **Multimedia items** | ✔ | ✔ | **6** |

## 2 RELATED WORK

**Visual Question Answering (VQA) Benchmarks.** Early VQA benchmarks focused on single-instance tasks such as image–text retrieval (Lin et al., 2015; Plummer et al., 2015; Goyal et al., 2017; Marino et al., 2019; Mathew et al., 2021), overlooking retrieval across heterogeneous sources. Later datasets (e.g., A-OKVQA (Schwenk et al., 2022), TextVQA (Singh et al., 2019), DocVQA (Mathew et al., 2021)) added external knowledge or documents but still assumed resolution from a single instance. Recent suites such as MM-Bench (Liu et al., 2024), MV-Bench (Li et al., 2024a), and OmniBench (Li et al., 2025) extend to video and audio, yet mainly test *intra-instance* understanding, not retrieval across instances. These settings overestimate model reliability, whereas real applications require retrieval with evidence-grounded reasoning, motivating **Needle-in-a-Haystack** evaluations.

**Needle-in-a-Haystack Benchmarks.** Recent efforts enlarge candidate pools to mimic real-world search: WebVQA (Doe & Smith, 2023) and RetVQA (Penamakuri et al., 2023) cover dozens of images; DocHaystack (Chen et al., 2024) scales to hundreds of document pages; MM-NIAH (Wang et al., 2024b) and MMNeedle (Wang et al., 2025) introduce mixed text–image or patch-level inputs. Despite progress, they face three key limitations: (1) single-modality focus; (2) limited cross-modal reasoning; and (3) ambiguous questions with multiple answers.

In contrast, MultiHaystack is, to our knowledge, the first benchmark to jointly consider large-scale retrieval, cross-modal integration, and unambiguous questions. Rather than offering incremental extensions within a single modality or small candidate pool, it explicitly targets the combination that prior work has left open, situating retrieval as the key challenge for realistic multimodal reasoning.

## 3 MULTIHAYSTACK

In this section, we present **MultiHaystack**, a large-scale benchmark to evaluate MLLMs on realistic cross-modal retrieval and reasoning. It is constructed through a four-stage pipeline including data collection, question generation, multi-step filtering, and enrichment to increase retrieval difficulty. The resulting corpus $\mathcal{D} = \{d_1, \ldots, d_N\}$ serves as the heterogeneous evidence pool, where each $d_i$ denotes a candidate item. Specifically, $\mathcal{D}$ spans three modalities: images, videos, and documents, as illustrated in Figure 4. Each question is constructed such that exactly one $d_i \in \mathcal{D}$ provides the uniquely supporting evidence, ensuring that answers are explicitly grounded and verifiable.

### 3.1 TASK DEFINITION

Each question in MultiHaystack first requires retrieving its supporting item from the multimodal heterogeneous corpus $\mathcal{D}$, and then performing fine-grained reasoning on it. This setup reflects realistic workflows where retrieval quality directly affects reasoning, so evaluation captures both stages together rather than in isolation.

### 3.2 DATA STATISTICS

**Comparison.** Prior benchmarks often restrict to a single modality, allow ambiguous queries without unique answers, or require no real retrieval beyond snippets (Figure 1). MultiHaystack addresses these issues by ensuring each query is evidence-grounded with exactly one supporting item in the corpus, requiring models to both retrieve and reason over this evidence. As summarized in Table 1,

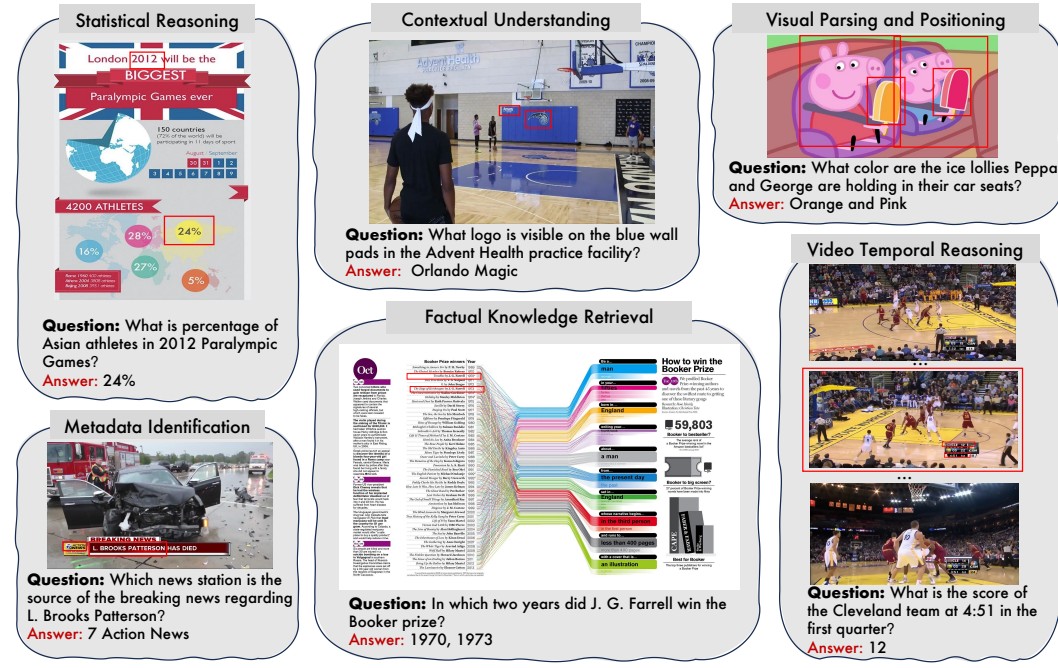

Figure 3: **Examples of six tasks in MultiHaystack:** Visual Parsing & Positioning (spatial layouts), Contextual Understanding (embedded text), Video Temporal Reasoning (motion/order), Statistical Reasoning (charts/tables), Metadata Identification (affiliations/timestamps), and Factual Knowledge Retrieval (corpus-grounded facts).

MultiHaystack is the first benchmark to scale across documents, images, and videos, while spanning six task types and integrating multi-needle retrieval with uniquely verifiable answers.

**Task Distribution.** The benchmark defines six tasks to evaluate MLLMs after retrieval from a large cross-modal corpus (Figure 3):

- **Visual Parsing and Positioning (VPP):** object positions and spatial layouts.
- **Contextual Understanding (CU):** embedded text in images or videos.
- **Video Temporal Reasoning (VTR):** motion, order, and state changes across frames.
- **Statistical Reasoning (SR):** quantitative patterns in charts, tables, and figures.
- **Metadata Identification (MI):** affiliations, timestamps, and institutions.
- **Factual Knowledge Retrieval (FKR):** corpus-grounded factual information.

Based on this design, MultiHaystack contains 33 visual parsing tasks, 30 contextual understanding tasks, 44 video temporal reasoning tasks, 321 statistical reasoning tasks, 285 metadata identification tasks, and 34 factual knowledge retrieval tasks. Each question is formed by inserting a controlled "needle," a uniquely answerable information fragment, into documents, images, or videos. This provides explicit semantic grounding and creates non-trivial retrieval challenges. Unlike prior benchmarks with ambiguous queries (Chen et al., 2024; Wang et al., 2025), **MultiHaystack** enforces unique and verifiable answers for rigorous, reproducible evaluation.

### 3.3 MULTIHAYSTACK CONSTRUCTION

To ensure coverage and uniquely verifiable answers, MultiHaystack is built through a four-stage pipeline: data collection, question generation, filtering, and enrichment (Figure 4).

**Stage 1: Data Collection.** We construct $\mathcal{D}$ by combining datasets from three modalities: videos (VideoVista (Li et al., 2024b), MMBench-Video (Fang et al., 2024), FineVideo (Farré et al., 2024), MVBench (Li et al., 2024a)), images (DocHaystack (Chen et al., 2024), MMIU (Meng et al., 2025), A-OKVQA (Schwenk et al., 2022)), and documents (MINT1T (Awadalla et al., 2024)). This curated mix brings cross-modal diversity beyond what any individual dataset offers.

Figure 4: **Benchmark construction pipeline.** MultiHaystack is built in four stages: collecting diverse multimodal sources, generating specific QA pairs, filtering for unique and grounded answers, and enriching with challenging distractors. This design ensures coverage across six tasks (Figure 3) and overcomes the unimodal, small-scale, or ambiguous limitations of prior benchmarks.

**Stage 2: Question generation.** Each file $d_i$ is first converted into images: PDF pages are rendered one by one, standalone images are used directly, and videos are uniformly sampled into eight frames. Based on these images, GPT-4o generates a set of QA pairs $Q_i$. On average, each item contributes around 30 candidate questions, forming the raw QA pool $\mathcal{Q} = \bigcup_i Q_i$.

**Stage 3: Question filtering.** We apply a three-step process to guarantee specificity and grounding. (1) GPT-4o and Gemini-2.5-Flash remove ambiguous questions with multiple valid answers. (2) Manual review discards items without explicit anchors such as objects, locations, or timestamps. (3) A retrieval-independence test eliminates questions solvable without consulting the supporting item. The resulting set $\mathcal{Q}^\star$ contains 747 questions linked to 433 images, 105 videos, and 209 documents (282 items total), balanced across modalities for fair evaluation.

**Stage 4: Data enrichment.** To increase retrieval difficulty, we construct distractors $\mathcal{D}^-$ for each question $q \in \mathcal{Q}^\star$. Candidates are first retrieved using GPT-4o keywords, then filtered by CLIP similarity and retriever scores to ensure semantic plausibility without redundancy. Manual verification further confirms that distractors never contain the correct answer, yielding a challenging yet unambiguous dataset of about 46K items.

**Manual verification.** Each query is validated with retrieval models and human checks to confirm that answers come only from the designated supporting item. Although annotations are defined at the page or frame level, evaluation is conducted at the item level, since models process whole documents or videos. This dual validation guarantees reliability and prevents shortcuts.

**Data profile.** The final benchmark includes 747 questions and 46,260 items: 25,652 images, 10,419 videos, and 10,189 documents. Additional examples and analyses are provided in Appendix B.

## 4 EXPERIMENTS

### 4.1 EXPERIMENTAL SETUP

**Methods.** To evaluate models on our benchmark, we follow retrieval-augmented pipelines (Chen et al., 2024) and adapt them by unifying input modalities: images are used directly, videos are represented by eight uniformly sampled frames, and documents are rendered into sequential images (Ma et al., 2024). QA pairs are annotated at the *page/frame level*, while retrieval is evaluated at the *item level*. A retrieval is considered correct if the file containing the annotated page or frame appears in the top-$k$ results. For the VQA stage, the full retrieved item is provided as input, closely simulating real-world retrieval-augmented generation. Vision–Language Models (VLMs), such as SigLIP (Zhai et al., 2023), are used to rank candidates; the top-$k$ are paired with the question and passed to the multimodal model for answer generation. When models impose input restrictions (e.g., limited sequence length or incomplete video support), we apply additional preprocessing to ensure compatibility.

Table 2: **Retrieval performance in cross-modality vs. single-modality settings.** Cross-modality results are shown in black, while single-modality results are shown in gray for comparison. Best values per column are highlighted in bold.

| Model | Video | | | Image | | | Document | | | Overall | | |
|---|---|---|---|---|---|---|---|---|---|---|---|---|
| | R@1 | R@3 | R@5 | R@1 | R@3 | R@5 | R@1 | R@3 | R@5 | R@1 | R@3 | R@5 |
| CLIP | 26.67 (56.19) | 40.00 (78.10) | 51.43 (80.00) | 21.71 (30.25) | 31.64 (40.88) | 34.87 (44.34) | 34.93 (38.76) | 46.89 (51.67) | 48.80 (53.59) | 26.10 (36.28) | 37.08 (49.13) | 41.10 (51.94) |
| SigLIP2 | **40.00** (63.81) | **60.00** (83.81) | **74.29** (91.43) | 32.10 (44.11) | 40.88 (53.12) | 45.27 (58.66) | **59.81** (61.72) | 68.42 (70.33) | 72.73 (75.12) | **40.96** (51.81) | 51.27 (62.25) | 57.03 (67.87) |
| OpenCLIP | 38.10 (60.00) | 56.19 (74.29) | 62.86 (78.10) | 19.40 (25.40) | 27.94 (35.80) | 32.33 (42.26) | 28.71 (32.06) | 36.84 (42.58) | 43.06 (47.85) | 24.63 (32.13) | 34.40 (43.11) | 39.63 (48.86) |
| Jina-Clip-V1 | 21.90 (42.86) | 38.10 (59.05) | 47.62 (67.62) | 7.39 (13.39) | 10.16 (19.17) | 12.93 (22.40) | 16.75 (17.70) | 21.05 (22.01) | 22.49 (22.97) | 12.05 (18.74) | 17.14 (25.57) | 20.48 (28.92) |
| Jina-Clip-V2 | 20.00 (36.19) | 30.48 (56.19) | 35.24 (76.19) | 11.78 (27.25) | 21.02 (42.73) | 25.17 (48.04) | 40.67 (41.63) | 51.67 (52.63) | 55.98 (56.46) | 21.02 (32.53) | 30.92 (47.39) | 35.21 (54.35) |
| NEV | 25.71 (38.10) | 40.00 (54.29) | 42.86 (60.95) | 5.31 (8.78) | 7.39 (12.01) | 8.78 (13.63) | 9.09 (10.53) | 12.92 (13.88) | 13.88 (16.27) | 9.24 (13.39) | 13.52 (18.47) | 14.99 (21.02) |
| E5-V | 34.29 (62.86) | 51.43 (81.90) | 60.95 (83.81) | **33.49** (43.19) | **55.20** (68.36) | **62.82** (73.44) | 59.33 (60.77) | **70.33** (71.29) | **75.12** (76.08) | 40.83 (50.87) | **58.90** (71.08) | **66.00** (75.64) |
| MM-Embed | 37.14 (60.95) | 47.62 (80.00) | 55.24 (87.62) | 31.41 (43.65) | 43.65 (64.43) | 51.27 (67.21) | 53.59 (62.68) | 62.68 (67.46) | 70.81 (75.60) | 38.42 (51.41) | 49.53 (67.47) | 57.30 (72.42) |

**Baselines.** We evaluate two categories of baselines: VLMs for retrieval and MLLMs for VQA.

For retrieval, we benchmark two categories of VLMs: *CLIP-based models*, including CLIP (Radford et al., 2021), OpenCLIP (Cherti et al., 2022), and Jina-CLIP v1/v2 (Koukounas et al., 2024a;b); and *multimodal embedding models*, including SigLIP2 (Tschannen et al., 2025), Nomic-Embed-Vision (Nussbaum et al., 2024), E5-V (Jiang et al., 2024), and MM-Embed (Lin et al., 2025).

For VQA, we evaluate two categories of MLLMs: *open-source models*, including Ola-7B (Liu et al., 2025), InternVL-3-8B (Zhu et al., 2025), and Qwen2-VL-7B (Wang et al., 2024a); and *proprietary models*, including GPT-5 (OpenAI, 2025) and Gemini-2.5-Flash (AI, 2024).

**Metrics.** Retrieval is measured at the *item level*, reporting Recall@1/3/5 to indicate whether the ground-truth item appears in the top-$k$ results, since real-world applications often require retrieving the entire file rather than a single page or frame. While long documents and videos are internally segmented into pages or frames to support downstream reasoning, retrieval always operates on the complete item, and evaluation strictly follows this item-level definition. VQA accuracy is evaluated using GPT-4o-mini under a fixed rubric, with manual auditing confirming high agreement with human judgment. Details of the rubric are provided in Appendix C.

## 4.2 RETRIEVAL RESULTS

Table 2 compares both single-modality and cross-modality retrieval to reveal the strengths and weaknesses of current retrieval models.

**Single Modality.** When restricted to a single modality, current models achieve strong performance. For instance, SigLIP2 exceeds 90% Recall@5 on videos, while MM-Embed surpasses 75% Recall@5 on documents. These results indicate that unimodal retrieval is near-saturated, and single-modality benchmarks no longer expose meaningful weaknesses.

**Cross Modalities.** In contrast, cross-modal retrieval remains highly challenging. Even the strongest models, SigLIP2 and E5-V, reach only 40.96% and 40.83% R@1—drops of over 40 points from their unimodal results. MM-Embed attains relatively higher recall at R@5 (57.30%), yet still falls well short of its unimodal performance. Weaker baselines degrade even further, with document retrieval proving the most difficult.

These findings underscore that cross-modal retrieval is the primary bottleneck for MLLMs, underscoring the need for benchmarks like MultiHaystack to measure and advance this capability.

## 4.3 VISUAL QUESTION ANSWERING RESULTS

As shown in Table 3, we compare VQA accuracy when conditioning on items retrieved from single-modality versus cross-modality search. GPT-5 achieves the highest overall performance, reaching 59.84% with single-modality retrieval but only 51.41% under cross-modal retrieval. Gemini-2.5-Flash follows a similar pattern, dropping from 50.87% to 43.64%. In contrast, weaker models such as Ola and Qwen2-VL remain below 25% overall even with unimodal

Table 3: **VQA performance.** Each model answers questions using top-5 items retrieved by E5-V from cross-modality inputs; gray numbers show single-modality Recall@5 for reference.

| Model | Video | Image | Document | Overall |
|---|---|---|---|---|
| Ola | 14.29 (22.86) | 20.09 (31.41) | 36.36 (44.94) | 23.83 (34.00) |
| InternVL-3 | 17.14 (20.95) | 29.33 (38.80) | 49.28 (51.67) | 33.29 (39.89) |
| Qwen2-VL | 16.19 (18.10) | 16.86 (24.94) | 19.62 (22.49) | 17.54 (23.29) |
| Gemini-2.5-Flash | 52.38 (61.90) | 35.10 (44.57) | 56.94 (58.37) | 43.64 (50.87) |
| GPT-5 | **60.00** (67.62) | **43.19** (52.66) | **64.11** (70.81) | **51.41** (59.84) |

retrieval, indicating limited grounding ability. These results demonstrate that retrieval errors propagate directly into reasoning, underscoring the importance of robust cross-modal grounding. Together

Table 4: **Retrieval results across six tasks.** Recall@1/3/5 of different vision-language retrieval models on MultiHaystack across six distinct tasks.

| Model | VPP | | | CU | | | VTR | | | FKR | | | SR | | | MI | | |
|---|---|---|---|---|---|---|---|---|---|---|---|---|---|---|---|---|---|---|
| | R@1 | R@3 | R@5 | R@1 | R@3 | R@5 | R@1 | R@3 | R@5 | R@1 | R@3 | R@5 | R@1 | R@3 | R@5 | R@1 | R@3 | R@5 |
| CLIP | 33.33 | 39.39 | 42.42 | 16.67 | 30.00 | 43.33 | 29.55 | 40.91 | 50.00 | 20.59 | 23.53 | 29.41 | 25.23 | 35.51 | 38.63 | 27.37 | 40.35 | 43.51 |
| SigLIP2 | **42.42** | **66.67** | **72.73** | **53.33** | 66.67 | **80.00** | 29.55 | 52.27 | 65.91 | **26.47** | 32.35 | 47.06 | 38.01 | 45.48 | 50.47 | **46.32** | 56.49 | 60.00 |
| OpenCLIP | 36.36 | 51.52 | 54.55 | 30.00 | 46.67 | 50.00 | **38.64** | **61.36** | 70.45 | 17.65 | 20.59 | 20.59 | 22.74 | 31.78 | 37.69 | 23.51 | 31.58 | 36.49 |
| Jina-Clip-V1 | 21.21 | 27.27 | 39.39 | 13.33 | 13.33 | 20.00 | 29.55 | 59.09 | 70.45 | 2.94 | 8.82 | 8.82 | 9.03 | 13.08 | 15.26 | 12.63 | 15.44 | 17.89 |
| Jina-Clip-V2 | 33.33 | 42.42 | 48.48 | 10.00 | 16.67 | 23.33 | 11.36 | 20.45 | 25.00 | 11.76 | 20.59 | 23.53 | 17.13 | 28.35 | 31.78 | 27.72 | 36.84 | 41.75 |
| NEV | 15.15 | 24.24 | 27.27 | 16.67 | 26.67 | 30.00 | 34.09 | 50.00 | 54.55 | 0.00 | 0.00 | 2.94 | 7.17 | 10.59 | 11.84 | 7.37 | 10.18 | 10.88 |
| E5-V | 42.42 | 57.58 | 66.67 | 36.67 | 43.33 | 43.33 | **38.64** | 61.36 | 70.45 | **26.47** | **44.12** | **58.82** | 38.63 | **62.31** | **69.78** | 45.61 | **58.25** | **64.21** |
| MM-Embed | 36.36 | 45.45 | 54.55 | 30.00 | 36.67 | 40.00 | 27.27 | 56.82 | 59.09 | 20.59 | 29.41 | 44.12 | 37.69 | 53.27 | 64.49 | 44.21 | 48.42 | 52.63 |

with the retrieval results, they demonstrate that progress in multimodal reasoning is inseparable from advances in cross-modal retrieval.

## 4.4 DISCUSSION

### 4.4.1 HOW DOES PERFORMANCE VARY ACROSS TASKS?

For reference, the aggregated results associated with the task-level analyses discussed here are already reflected in the "Overall" columns of Tables 2 and 3, which summarize the results of Tables 4 and 5.

**Retrieval.** Table 4 shows that retrieval performance varies widely across tasks, highlighting distinct capability gaps. SigLIP2 and E5-V achieve over 60% Recall@5 on Visual Parsing and Positioning as well as Video Temporal Reasoning, indicating strength in spatial parsing and temporal understanding. However, both drop below 50% on Factual Knowledge Retrieval and Statistical Reasoning, highlighting factual grounding and quantitative reasoning as major weaknesses. MM-Embed is more balanced across tasks, but its advantage does not close the gap on reasoning-intensive categories. These discrepancies show that aggregate retrieval metrics can mask meaningful weaknesses, and our task-level categorization enables more precise diagnosis of where models fail.

**Visual Question Answering.** Table 5 shows that reasoning performance closely mirrors retrieval difficulties. GPT-5 achieves the best overall results, with strong accuracy on Metadata Identification (58.95%) and Visual Parsing and Positioning (57.58%). However, its accuracy drops sharply on Statistical Reasoning (43.61%), indicating that numerical reasoning remains a

Table 5: **Comparison of MLLMs' VQA performance integrated with E5-V across six tasks.**

| Model | VPP | CU | VTR | FKR | SR | MI |
|---|---|---|---|---|---|---|
| Ola | 27.27 | 30.00 | 18.18 | 20.59 | 26.17 | 21.40 |
| InternVL-3 | 42.42 | 23.33 | 11.36 | 23.53 | 29.91 | 41.40 |
| Qwen2-VL | 18.18 | 23.33 | 6.82 | 14.71 | 18.38 | 17.89 |
| Gemini-2.5-Flash | 54.55 | 46.67 | 56.82 | 32.35 | 35.51 | 50.53 |
| GPT-5 | **57.58** | **56.67** | 52.27 | **50.00** | **43.61** | **58.95** |

bottleneck. Gemini-2.5-Flash follows a similar trend, while weaker models such as Ola and Qwen2-VL remain below 30% on most tasks, even with retrieved context. These findings highlight that multimodal reasoning ability is uneven across task types, making fine-grained evaluation indispensable. By disentangling task categories, our benchmark not only stresses retrieval at scale but also reveals the reasoning dimensions where progress is most needed.

### 4.4.2 HOW DOES TOP-k AFFECT RETRIEVAL AND REASONING?

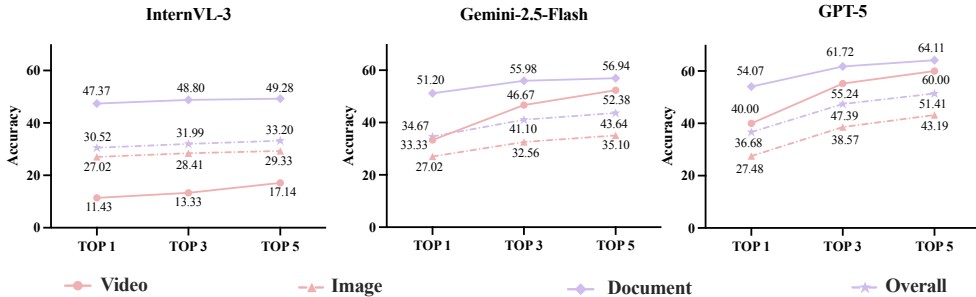

Figure 5: **Top-$k$ ablation analysis for vision–language models integrated with E5-V.**

Figure 5 presents a top-$k$ ablation analysis for three vision–language models. As expected, VQA accuracy improves when moving from Top-1 to Top-5 retrieved items, further confirming that retrieval

coverage is a critical bottleneck. GPT-5 benefits the most, reaching 64.11% overall accuracy at Top-5, while Gemini-2.5-Flash shows moderate gains, and InternVL-3 remains consistently weaker. Across modalities, documents yield the largest improvements, whereas video and image retrieval provide smaller but steady gains. These results highlight that access to a broader candidate set can mitigate retrieval errors, yet reasoning robustness is equally essential: even with five retrieved items, substantial gaps remain across modalities and models.

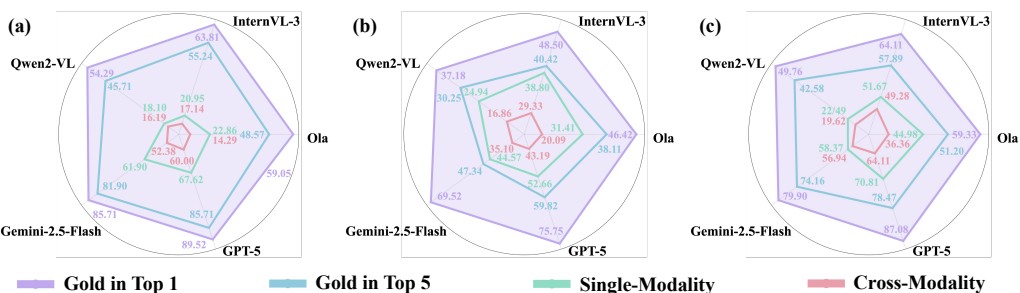

Figure 6: **Performance comparison in three distinct modalities.** (a) represents the video modality, (b) represents the image modality, and (c) represents the document modality.

### 4.4.3 WHAT IS THE GAP BETWEEN GOLD, SINGLE-MODALITY, AND CROSS-MODALITY SETTINGS?

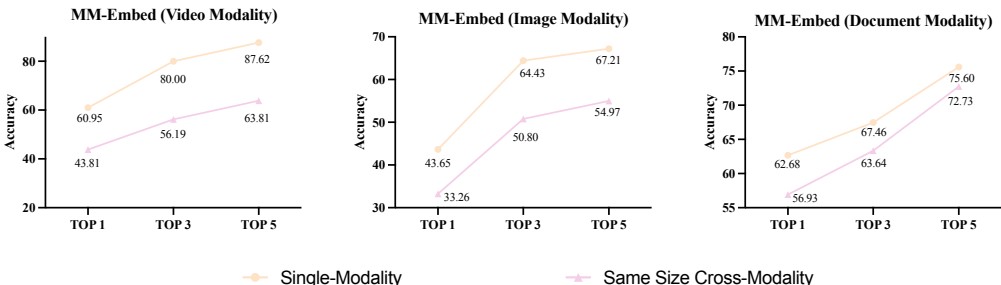

Figure 7: **Pool-size controlled comparison.** Accuracy of MM-Embed under single-modality retrieval and mixed-modality retrieval with an identical total pool size. Performance remains substantially lower in the mixed-modality condition, indicating that the cross-modal gap arises from modality heterogeneity rather than pool size.

Figure 6 contrasts model performance under Gold, single-modality, and cross-modality settings across video, image, and document tasks. Gold Top-1/5 serves as the upper bound, where models are directly provided with the answer-containing file and achieve the highest accuracy. Single-modality retrieval approaches this level in all three cases, suggesting that current models appear reliable when evidence stays within one modality. However, cross-modality retrieval leads to substantial declines, with the largest gap observed in video, followed by documents and then images. Even GPT-5 suffers pronounced drops once heterogeneous modalities appear, indicating that unimodal evaluations conceal the weaknesses that matter most in realistic multimodal scenarios.

To further understand the source of this degradation, Figure 7 compares single-modality and mixed-modality retrieval under the same total pool size, providing a controlled test of whether the decline stems from candidate set size or modality heterogeneity. Performance remains consistently lower in the mixed-modality case, demonstrating that the difficulty is not due to pool size but to cross-modal embedding mismatch and semantic interference. Together, these trends reinforce the benchmark's motivation: reasoning is generally reliable once the correct evidence is surfaced, yet identifying the correct evidence in large heterogeneous environments remains the dominant bottleneck.

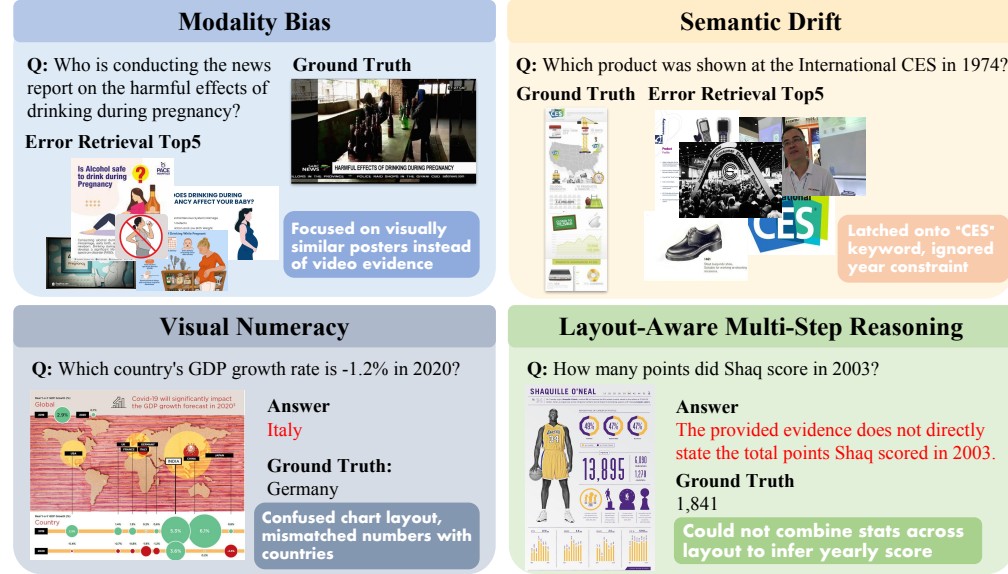

Figure 8: **Representative error cases** illustrating retrieval errors such as *modality bias* (retrieving images instead of video evidence) and *semantic drift* (ignoring temporal constraints), as well as reasoning errors such as *visual numeracy* (mismatching numbers in charts) and *layout-aware multi-step reasoning* (failing to combine structured cues across layouts).

### 4.4.4 CAN LLMs SERVE AS RELIABLE JUDGES?

We sampled 30 non-overlapping QA pairs for each model and asked MTurk workers to label answers as correct or incorrect, then compared their judgments with those of GPT-4o-mini. Table 6 exhibits strong consistency between GPT-4o-mini and human annotations:

Table 6: **Reliability Matrix for LLM-as-Judge**

| Model | Cohen's $\kappa$ | 95% CI | Accuracy | Pearson $r$ |
|---|---|---|---|---|
| Ola | 0.918 | [0.710, 1.000] | 0.967 | 0.921 |
| InternVL-3 | 0.865 | [0.667, 1.000] | 0.933 | 0.873 |
| Qwen2-VL | 1.000 | [1.000, 1.000] | 1.000 | 1.000 |
| Gemini-2.5-Flash | 0.932 | [0.772, 1.000] | 0.967 | 0.934 |
| GPT-5 | 1.000 | [1.000, 1.000] | 1.000 | 1.000 |

Cohen's $\kappa$ exceeds 0.86 and accuracy remains above 93% across all models, with Qwen2-VL and GPT-5 reaching perfect agreement. Pearson correlations are near 1.0 for the remaining models. These results indicate that GPT-4o-mini achieves human-level reliability in answer verification, enabling evaluation that is rigorous, reproducible, and cost-efficient.

### 4.4.5 WHY IS DATA ENRICHMENT ESSENTIAL FOR LARGE-SCALE EVALUATION?

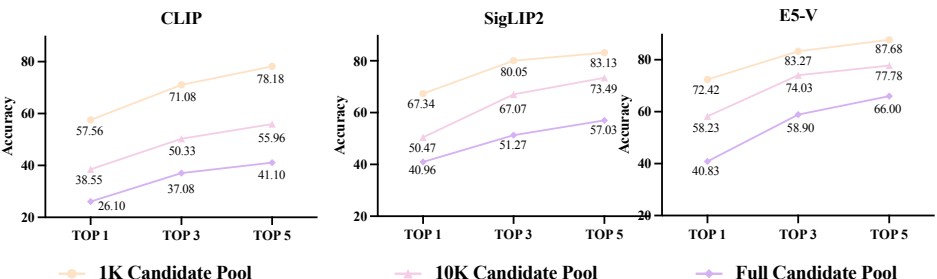

Figure 9: **Effect of data enrichment under varying candidate pool sizes,** showing that accuracy consistently drops as the pool expands.

We simulate realistic retrieval conditions by enriching candidate pools: all positives are included, and distractors are progressively added until the pool reaches 1K, 10K, and the full corpus. Under this setup, systems must identify relevant information within massive and noisy collections. As shown in Figure 9, performance declines as the pool expands, and the rate of degradation reveals the robustness of different models. Models like CLIP suffer steep drops, indicating over-reliance on surface-level similarities, while SigLIP2 and E5-V degrade more gradually, reflecting stronger discriminative ability at scale. Crucially, such contrasting robustness would remain hidden in small-scale evaluations. By exposing a large performance gap between small-scale and full candidate pools, data enrichment proves essential for any benchmark that aims to faithfully reflect real-world large-scale retrieval.

Table 7: **Preliminary studies with advanced retrieval pipelines.**

| Methods | Video | | | Image | | | Document | | | Overall | | |
|---|---|---|---|---|---|---|---|---|---|---|---|---|
| | R@1 | R@3 | R@5 | R@1 | R@3 | R@5 | R@1 | R@3 | R@5 | R@1 | R@3 | R@5 |
| *Single-Modality* | | | | | | | | | | | | |
| E5-V | 62.86 | 81.90 | 83.81 | 43.19 | 68.36 | 73.44 | 60.77 | 71.29 | 76.08 | 50.87 | 71.08 | 75.64 |
| *Cross-Modality* | | | | | | | | | | | | |
| E5-V | 34.29 | 51.43 | 60.95 | 33.49 | 55.20 | 62.82 | 59.33 | 70.33 | 75.12 | 40.83 | 58.90 | 66.00 |
| E5-V + Refined Query | 41.90 | 59.05 | 64.76 | 36.26 | 56.58 | 66.74 | 60.29 | 70.81 | 75.12 | 43.78 | 60.91 | 68.81 |
| E5-V + MMSearch (Jiang et al., 2025) | 44.76 | 62.86 | 65.71 | 36.95 | 57.97 | 69.52 | 60.29 | 72.73 | 75.60 | 44.58 | 62.78 | 70.68 |
| VisRAG (Yu et al., 2025) | 40.95 | 64.76 | 68.57 | 39.03 | 45.96 | 50.12 | 60.77 | 67.46 | 70.33 | 45.38 | 54.62 | 58.37 |

## 5 ERROR ANALYSIS

This section provides a detailed error analysis (Figure 8), distinguishing between retrieval failures (locating relevant evidence) and reasoning failures (interpreting the retrieved evidence).

**Retrieval errors.** VLMs exhibit *modality bias*, retrieving images over video evidence, and *semantic drift*, favoring frequent entities while ignoring temporal constraints, leading to failures in temporal or factual queries. These errors reveal a misalignment between retrieval objectives and query intent, underscoring the need for models with modality awareness and constraint reasoning.

**Reasoning errors.** Even with correct evidence, MLLMs struggle with *visual numeracy*, often mismatching numbers with chart labels, and with *layout-aware multi-step reasoning*, failing to combine structured cues across layouts, leading to errors in tasks requiring precise multi-hop inference. These limitations point to gaps in fine-grained perceptual grounding and compositional reasoning, suggesting that stronger numeracy modules and layout-sensitive architectures are needed.

## 6 FUTURE DIRECTIONS

Table 7 reports early experiments with refined-query RAG, VisRAG (Yu et al., 2025), and representative agentic systems, MMSearch (Jiang et al., 2025). Although these pipelines incorporate improved query rewriting, iterative verification, and shallow re-ranking, they yield only modest gains over naïve retrieval and remain far below the single-modality upper bounds in Section 4. Under the E5-V cross-modal retrieval setting, this persistent gap indicates that the bottleneck does not lie in query formulation or superficial agent loops, but in fundamental challenges of cross-modal grounding: embedding mismatches across modalities, loss of fine-grained spatial or textual cues during compression, and the difficulty of integrating heterogeneous evidence with temporal or numerical structure.

These findings point to several research opportunities. More expressive and modality-aware cross-modal representations are needed to capture layout regularities in documents, temporal consistency in videos, and localized visual semantics in images, while still enabling comparable embeddings. Beyond representation learning, retrieval and reasoning must be more tightly coupled, allowing dynamic hypothesis refinement, adaptive query rewriting, and continuous re-ranking, rather than a static retrieval stage. Finally, systematic failures in contextual disambiguation, statistical reasoning, and fine-grained grounding highlight the need for architectures that combine structural priors with adaptive reasoning. By making these bottlenecks explicit and measurable, MultiHaystack offers a diagnostic foundation for evaluating and developing next-generation cross-modal RAG systems.

## 7 CONCLUSION

We introduced **MultiHaystack**, a large-scale benchmark for evaluating Multimodal Large Language Models in realistic cross-modal retrieval and reasoning. With over 46,000 images, videos, and documents paired with evidence-grounded questions, it systematically reveals limitations that single-modality evaluations conceal. Our results highlight the need for retrieval-aware reasoning and modality-agnostic architectures. We expect MultiHaystack to serve as a rigorous testbed for advancing multimodal intelligence, guiding future progress toward scalable and trustworthy real-world systems.

ETHICS STATEMENT

All data used in MultiHaystack are sourced from publicly available datasets with appropriate licenses or research consent. The benchmark construction pipeline included automated and manual filtering to remove ambiguous or sensitive content. No private information, personal identifiers, or confidential documents are involved. The benchmark is released strictly for research purposes to enable reproducible and transparent evaluation, and it does not pose privacy or security risks.

REPRODUCIBILITY STATEMENT

Upon acceptance of this paper at a peer-reviewed archival venue, we will release the MultiHaystack dataset. In the meantime, the supplementary materials include code for retrieval and evaluation, and we also provide several reference subsets of MultiHaystack for reference. Dataset statistics, the construction pipeline, and evaluation details are described in the main paper and Appendices A and D, and all baseline models are either open-sourced or publicly accessible. Together, these resources ensure that our results can be fully reproduced and fairly compared by the community.

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

APPENDIX

CONTENTS

# A STATISTICS

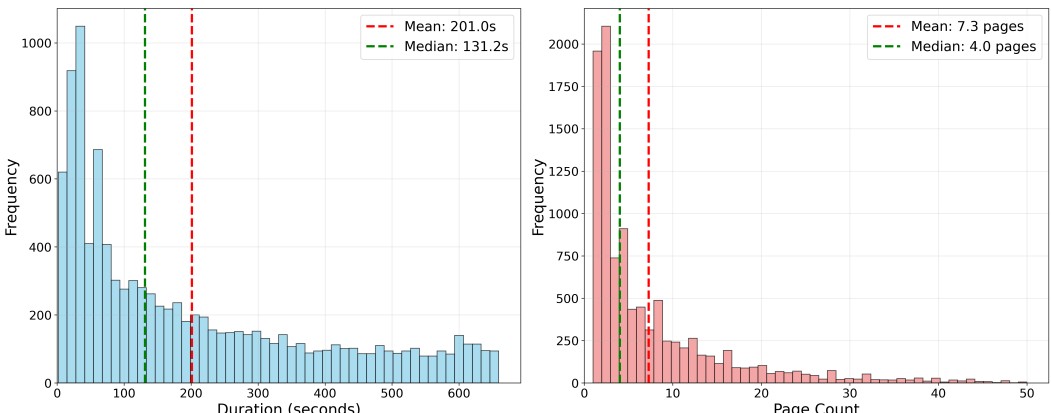

Figure A.1: **Video–Document Distribution Overview.** Distributions of video duration (left) and document page count (right), with red dashed lines indicating means and green dashed lines indicating medians.

Figure A.1 provides a corpus-level overview of sample lengths for two modalities in our benchmark: video (seconds) and document (pages), with means and medians annotated for reference. Both distributions are distinctly right-skewed, with many short items and a non-trivial long tail—statistics that mirror real-world multimedia collections and that are particularly relevant for retrieval under variable context sizes. This heterogeneity ensures that systems are evaluated on both rapid evidence localization in concise items and robust reasoning over extended content. The image modality comprises atomic, single-frame items and therefore has no analogous length measure. We report these statistics to characterize the benchmark and to contextualize evaluation difficulty, facilitating reproducibility and fair comparison across methods.

# B EXAMPLES FROM MULTIHAYSTACK

To illustrate the diverse and complex nature of the MultiHaystack benchmark, we present representative examples across video, image, and document modalities, including data-enriched cases. Each instance is designed for retrieval-augmented reasoning at scale, emphasizing both modality-specific understanding and fine-grained grounding.

These examples demonstrate that MultiHaystack provides a comprehensive and rigorous benchmark for cross-modal retrieval and reasoning, capturing both perceptual diversity and semantic nuance under realistic large-scale conditions.

## B.1 MODALITY EXAMPLES

### B.1.1 VIDEO

**Video-based QA** often requires modeling temporal dynamics, capturing frame-level details, and leveraging embedded textual cues. For instance, in Figure B.1, the model must detect a small facial accessory (a nose ring) while the subject applies hair dye, illustrating the need for fine-grained perceptual grounding under distracting context. Figure B.2 demands recognition of a brand logo in a low-resolution news segment, testing robustness to visual degradation. Figure B.3 evaluates domain-level inference from motion-blurred frames, where temporal context must compensate for reduced visual clarity. Together, these tasks highlight the dual challenges of temporal sensitivity and perceptual precision in video retrieval.

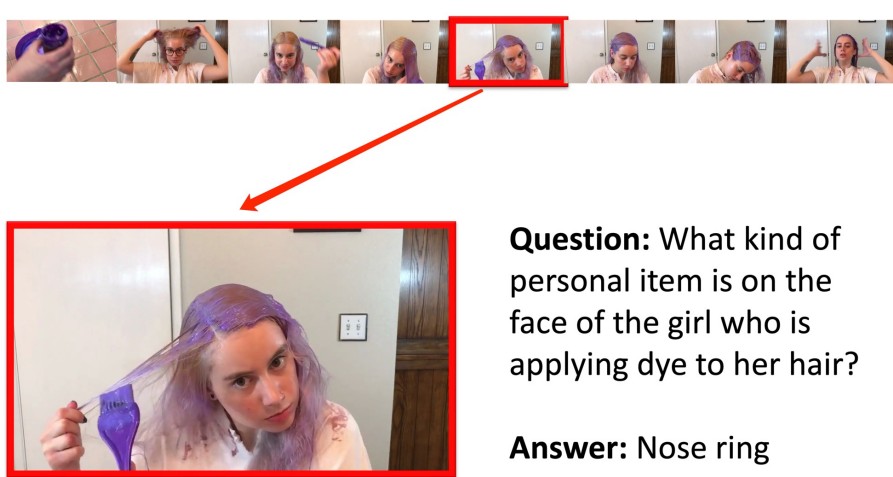

Figure B.1: **Video Example 1.**

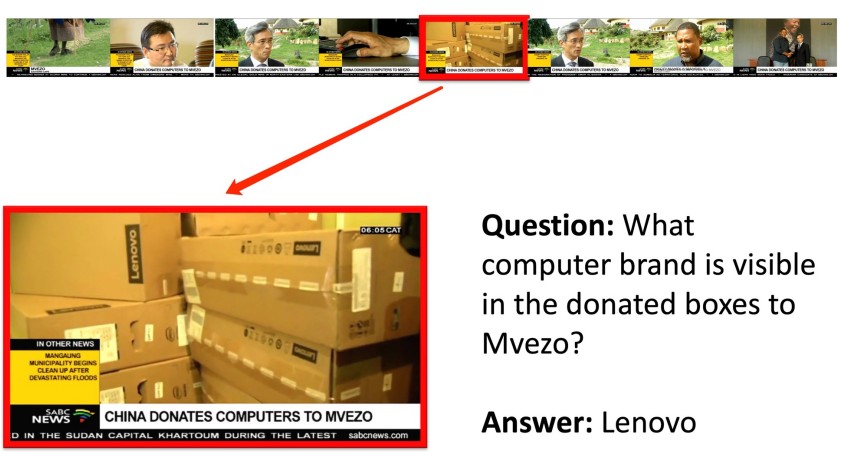

Figure B.2: **Video Example 2.**

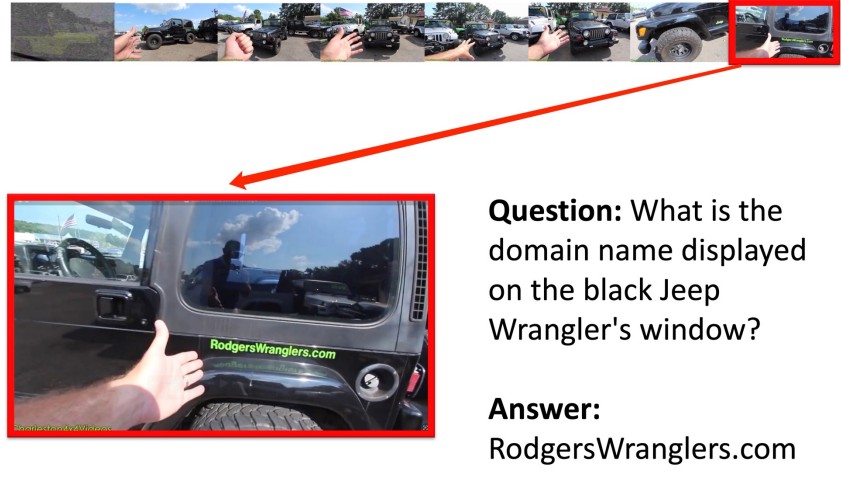

Figure B.3: **Video Example 3.**

### B.1.2 IMAGE

**Image-based QA** emphasizes spatial understanding and localized recognition. As shown in Figure B.4, the model must infer color attributes from real-world marketplace settings. Figure B.5 requires recognizing small object co-occurrence (a horse next to an apple), while Figure B.6 focuses on identifying object properties (a black bag in a laundry room). These examples test the model's capability to reason over everyday scenes with high visual clutter and subtle semantic cues.

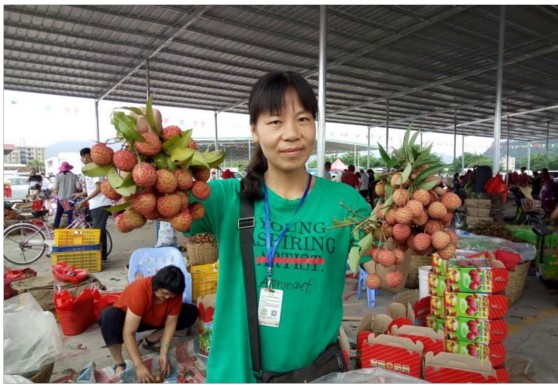

**Question:** What is the color of the clothes wears by the woman who holds Lychee?

**Answer:** Green

Figure B.4: **Image Example 1.**

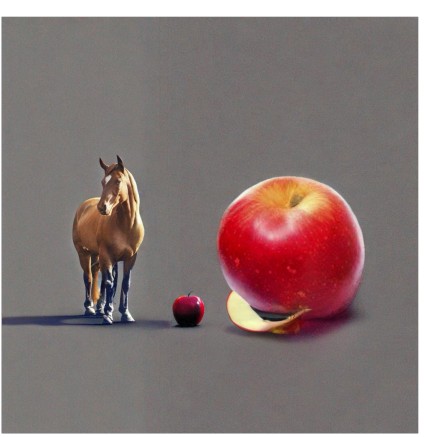

**Question:** What animal is put in the left side of the apple?

**Answer:** Horse

Figure B.5: **Image Example 2.**

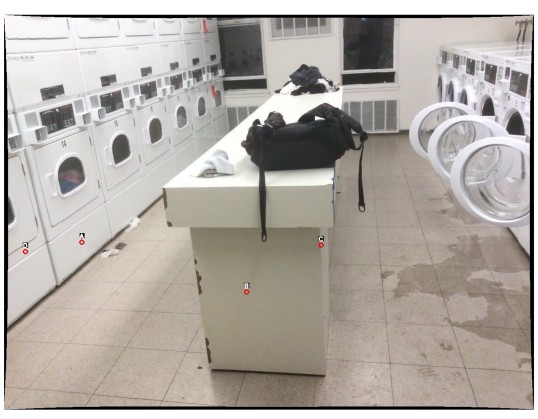

**Question:** What is the color of the bag that is put on the table in the laundry room?

**Answer:** Black

Figure B.6: **Image Example 3.**

### B.1.3 DOCUMENT

**Document-based QA** requires both visual–textual alignment and structured content reasoning. In Figure B.7, the model must locate and integrate a technical concept introduced jointly in scientific text and figures, demanding precise cross-modal grounding. Figure B.8 involves extracting factual content from narrative passages, testing robustness to linguistic variability and contextual dependencies. Figure B.9 requires retrieving quantitative results (e.g., mean average precision) from densely packed tables, highlighting the difficulty of parsing layout-dependent numerical data. Together, these tasks illustrate the need for accurate text extraction, layout-aware reasoning, and fine-grained multimodal understanding in document QA.

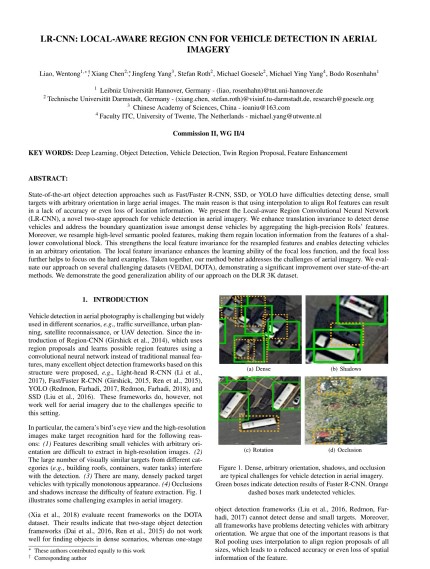

**Question:** What novel approach is introduced for vehicle detection in aerial imagery?

**Answer:** Local-aware Region Convolutional Neural Network (LR-CNN)

Figure B.7: **Document Example 1.**

**Question:** What surgery did Samantha Kinzalow undergo last Thursday?

**Answer:** Thyroid surgery

Figure B.8: **Document Example 2.**

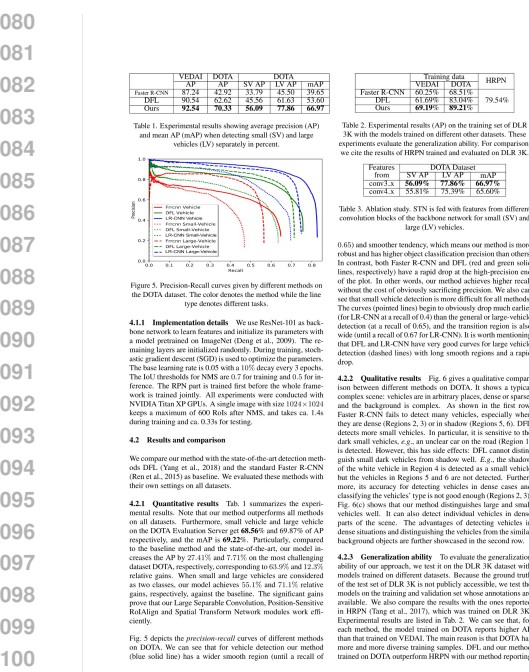

**Question:** What is the mean average precision (mAP) achieved by the DFL model for large vehicles in the DOTA dataset?

**Answer:** 53.60%

Figure B.9: **Document Example 3.**

## B.2 DATA ENRICHMENT EXAMPLES

In addition to ground-truth sources, MultiHaystack incorporates **data-enriched contrastive examples** that bear strong semantic or visual similarity to the correct content but do not contain the target answer. The inclusion of these examples is motivated by the need to reflect the inherent ambiguity present in real-world retrieval scenarios, where multiple plausible candidates often appear contextually relevant despite being incorrect. Rather than artificially introducing noise, these examples are carefully selected based on contextual coherence and fine-grained resemblance, ensuring that they remain informative and challenging. As illustrated in Figures B.10–B.12, these contrastive examples are constructed to simulate realistic retrieval confusion without relying on synthetic perturbations. For instance, Figure B.10 presents an electronics-related scene that is temporally close to the target reference but does not include the specified CES product. Figure B.11 shows a visually similar cartoon frame that lacks the queried object state. Figure B.12 depicts a relevant industrial setting, yet omits the specific label required by the question.

This design aligns closely with the task types defined in Figure 3 in the main text, particularly those requiring contextual understanding, visual parsing, and metadata identification. In these tasks, distinguishing semantically proximate yet incomplete candidates is essential for accurate reasoning. Moreover, such contrastive examples mirror the uncertainty faced in open-domain QA systems, where models must search over large corpora containing numerous partially relevant documents. By introducing semantically aligned but unanswerable instances, MultiHaystack encourages precise grounding and discourages superficial similarity matching, thereby offering a more faithful evaluation of retrieval and reasoning capabilities in realistic multimodal settings.

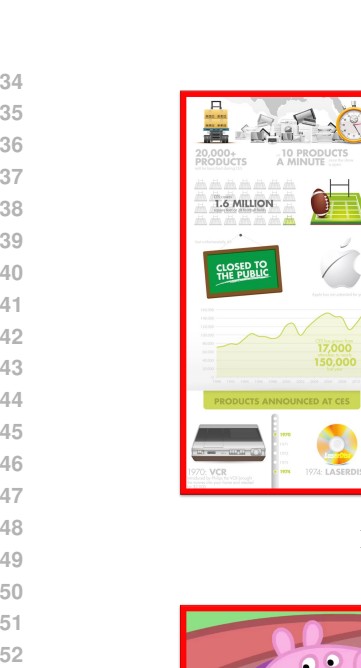
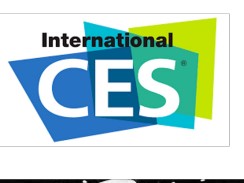
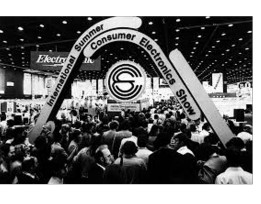

**Question:** Which product was shown at the International CES in 1974?

**Answer:** LASERDISC

Figure B.10: **Data Enrichment Example 1.**

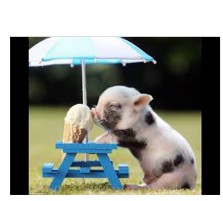
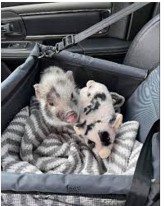

**Question:** What color are the ice lollies Peppa Pig and George Pig are holding in their car seats?

**Answer:** Orange and Pink

Figure B.11: **Data Enrichment Example 2.**

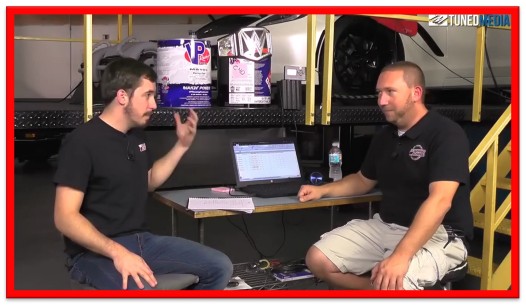
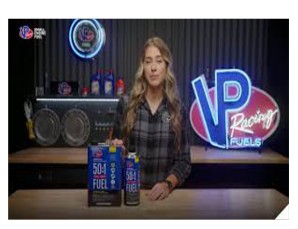
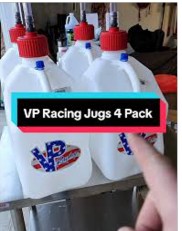

**Question:** What product model is visible on the VP Racing Fuels canister next to the laptop in the workshop setting?

**Answer:** MS109 Racing Fuel

Figure B.12: **Data Enrichment Example 3.**

## C  PROMPTS

In this section, we present the prompts for data construction and evaluation: one enforces precise, unambiguous QA generation, and the other defines a binary protocol for judging predictions, together ensuring reliable and reproducible assessment.

---

**QA Generation Prompt**

```
You are an expert at generating specific and precisely
targeted questions based on a series of sequentially provided
images.  These images have been sampled in sequence from a
video or a document and typically contain information-rich
visuals.
Your Task:
Carefully analyze all provided images.
Generate exactly 30 distinct, highly-specific questions,
each accompanied by its correct answer explicitly based on
information visible in these images.
Strict Requirements for Question Generation:
DO NOT reference the images themselves or use vague
positional indicators such as:
'in the image,' 'in the first/second/third image,' 'in the
provided picture,' 'at the top/bottom,' 'this slide,' 'the
table above/below,' etc.
Instead, always clearly and explicitly include specific
details directly from the visuals, such as:
Exact names (persons, companies, products)
Precise numerical values (financial figures, percentages)
Exact dates or years explicitly mentioned
Specific places, titles, labels, captions, or identifiable
entities clearly visible and named in the visuals.
Each question must stand alone as a fully self-contained,
specific question that would allow someone seeing it later to
precisely identify and locate the correct information within
related textual documents or sources, without needing visual
context.
Clear Examples of Correct Question Format (follow exactly
this style of specificity):
"What is the 'net earnings' of Johnson & Johnson and
subsidiaries in the year 2009?"
"What was the 'gross profit' reported by Johnson & Johnson
and subsidiaries for the fiscal year 2010?"
Explicitly Prohibited Example (Do NOT do this):
Incorrect:  "What are the two cats interacting with on
the wooden floor in the second image?" (Reason:  includes
prohibited phrase 'in the second image')
Corrected:  "What object are the two cats interacting
with on the wooden floor next to the white sofa?" (Reason:
explicitly references visual details, removing vague
positional indicators.)
Final Output Requirements:
Generate exactly 30 questions with their correct, concise
answers based explicitly on the details shown in the provided
series of images.
Ensure every question strictly follows the specificity
rules described above, completely eliminating unspecific or
ambiguous references to images or their positions.
```

---

**LLM Judgement Prompt**

```
You are an evaluator.  Compare the Predicted Answer with the
True Answer and determine if the Predicted Answer is Correct
or Incorrect.
Instructions:
1.  If the Predicted Answer provides the same information or
a reasonable interpretation of the True Answer, respond with
'Correct.'
2.  If the Predicted Answer does not match or does not
reasonably interpret the True Answer, respond with
'Incorrect.'
Important:  Answer only with 'Correct' or 'Incorrect' – no
explanations.
```

---

## D  REPRODUCIBILITY

### D.1  IMPLEMENTATION DETAILS

**Parameter settings** Across all experiments, the language model temperature was fixed at 0.4. During data enrichment, we employed CLIP-based filtering of web-retrieved images, retaining an image as a candidate distractor only if its cosine similarity with the corresponding query exceeded 0.2. For the VQA experiments, retrieval used a fixed top-k setting with k=5. In addition, the VQA pipeline implemented an automatic retry mechanism to improve robustness: if an error occurred at any stage, the procedure was retried up to three times before being marked as failed.

**Implementation Environment** All experiments were executed on a single NVIDIA H100 GPU (80 GB HBM3). The software stack comprised Python 3.12, PyTorch 2.6.0, and Hugging Face Transformers 4.51.0. Unless otherwise specified, inference was performed in bfloat16 (bf16) precision. These version details are reported to facilitate reproducibility.

**Task Distribution.** The task distribution in Figure 2 is deliberately designed rather than sampled from real-world frequencies. Our core motivation is to build a diagnostic benchmark: real-world distributions are long-tailed and dominated by easy perceptual queries, which severely under-represent harder reasoning types such as statistical analysis or metadata identification. If we followed such organic distributions, aggregate benchmark scores would largely reflect surface-level perception skills while masking weaknesses in deeper reasoning, thereby limiting the benchmark's value for research. To address this, we enforce a balanced coverage across six categories: (i) Visual Parsing and Positioning, targeting spatial localization and object layout; (ii) Contextual Understanding, focusing on embedded text and local semantics; (iii) Video Temporal Reasoning, requiring comprehension of motion and temporal order; (iv) Statistical Reasoning, evaluating quantitative analysis of tables and charts; (v) Metadata Identification, stressing recognition of affiliations, timestamps, and sources; and (vi) Factual Knowledge Retrieval, ensuring grounding in corpus-level factual evidence. These categories were carefully chosen to span perceptual and analytical dimensions, covering the dominant reasoning skills demanded in real-world multimodal applications. By balancing across them, the benchmark ensures fair and reproducible evaluation, highlights fine-grained strengths and weaknesses of models, and provides a controlled yet realistic setting to stress-test multimodal retrieval and reasoning capabilities.

### D.2  USAGE BENCHMARKS

- **VideoVista** (Li et al., 2024b) is a comprehensive video question answering benchmark with 24,906 multiple choice questions built from 3,402 YouTube videos across 14 categories, spanning a few seconds to over 10 minutes and covering 27 task types for understanding and reasoning. It is constructed via an automated pipeline that uses GPT-4o with video splitting, object segmentation, tracking, OCR, and ASR, followed by targeted human checks to ensure quality. Evaluations show persistent challenges in fine-grained temporal localization, anomaly detection, and relational and logical reasoning.

- **MMBench-Video** (Fang et al., 2024) is a long-form, multi-shot VideoQA benchmark designed to holistically assess LVLMs' spatial and temporal understanding across real-world web videos. It comprises 609 YouTube clips (30s–6min) spanning 16 categories and 1,998 human-authored, free-form QAs annotated under a 3-level taxonomy covering 26 fine-grained capabilities, with deliberate emphasis on temporal indispensability. The benchmark pairs open-ended evaluation with a GPT-4–based judging scheme to improve robustness and alignment with human preferences, and we report comprehensive comparisons of open-source and proprietary models. Code and evaluation are integrated into VLMEvalKit, providing a practical, scalable resource for advancing video understanding research.

- **FineVideo** (Farré et al., 2024) is a large-scale dataset for multimodal video understanding that targets the hard problems of mood analysis, narrative structure, and media editing. Spanning 43,751 YouTube videos ( 3,425 hours; avg. 4.7 minutes) across 122 categories, it couples raw video with time-coded speech-to-text and rich, scene-level annotations—characters, activities, props, editing cues, audiovisual correlation, narrative progression, and emotional trajectories. This fine-grained supervision enables both pretraining and task-specific fine-tuning for context-savvy video models.

- **MVBench** (Li et al., 2024a) is a comprehensive benchmark for temporal video understanding in MLLMs, defining 20 temporally grounded tasks by transforming static image tasks into their dynamic video counterparts. Multiple-choice questions are automatically generated from annotations across 11 public video datasets to ensure objective, reproducible scoring. Initial evaluations reveal considerable headroom for temporal reasoning, with the VideoChat2 baseline substantially outperforming prior models, establishing MVBench as a standardized, motion-aware testbed spanning perception through cognition.

- **DocHaystack** (Chen et al., 2024) is the large-scale benchmark for vision language reasoning that pairs each question with up to 1000 visual documents and requires a single document-grounded answer. Built from DocVQA and InfographicVQA using a pipeline that combines LLM filtering, human review, and removal of generic knowledge questions, they better reflect real retrieval needs at scale. The suite offers 100, 200, and 1000 document settings for joint evaluation of retrieval and VQA, with Recall at k used to assess retrieval quality.

- **MMIU** (Meng et al., 2025) is a comprehensive multi-image benchmark for evaluating large vision–language models, spanning 7 inter-image relationship types and 52 tasks built over 77,659 images and 11,698 carefully curated multiple-choice questions across five modalities, with an explicit unanswerable set for robustness analysis. Designed via a top-down hierarchy inspired by cognitive psychology, MMIU supports fine-grained diagnosis of semantic, temporal, and spatial reasoning, offers task-map analyses to distinguish in- vs. out-of-domain skills, and provides SFT-based difficulty estimates to guide model and data improvement.

- **A-OKVQA** (Schwenk et al., 2022) is a knowledge-intensive VQA benchmark built on COCO-2017 that comprises 24,903 question–answer–rationale triplets with train/val/test splits preserved, targeting reasoning that combines visual understanding with commonsense, factual, and physical world knowledge rather than simple lookup. Each item includes multiple-choice options and ten free-form answers, enabling both MC and Direct Answer evaluation, while human-written rationales (three per question) support training and analysis of explainable models. Compared with prior knowledge-based VQA datasets (e.g., OK-VQA), A-OKVQA is larger and uniquely provides sentence-level rationales, yielding a more diverse and challenging testbed for multimodal reasoning.

- **MINT1T** (Awadalla et al., 2024) is a large-scale open source multimodal interleaved dataset that preserves image and text order, assembled from HTML, PDFs, and arXiv at trillion token and billion image scale. It uses targeted quality filtering, NSFW screening, limited PII redaction, and extensive deduplication across text and images to improve cleanliness and diversity. Compared to OBELICS, it provides broader coverage with longer and more image-dense documents, and models trained on it achieve competitive or improved results on multimodal benchmarks.

## D.3 EVALUATION MODELS

- **CLIP** (Radford et al., 2021) is a dual-encoder vision–language model that aligns images and text in a shared embedding space via a symmetric contrastive objective over large batches. Trained on hundreds of millions of image–text pairs, it enables zero-shot recognition by turning class names or descriptions into text prompts that act as a classifier. This design yields strong, scalable performance across diverse benchmarks without task-specific fine-tuning.

- **SigLIP2** (Tschannen et al., 2025) is a multilingual vision and language encoder family that remains architecture-compatible with SigLIP and uses a unified training recipe combining a sigmoid image-text objective, a decoder for captioning and localization, and self-distillation with masked prediction to strengthen dense and spatial features; a NaFlex variant supports native aspect ratios and multiple resolutions, and the models deliver strong zero-shot classification and retrieval alongside improved localization and dense prediction.

- **OpenCLIP** (Cherti et al., 2022) is an open source CLIP training and evaluation stack built on LAION data that enables fully reproducible studies of scaling laws; trained on billions of image text pairs, it releases the largest public CLIP models and shows that the training distribution drives task-dependent scaling, with OpenCLIP improving more on zero-shot retrieval while OpenAI CLIP improves more on zero-shot classification, alongside strong results on ImageNet, VTAB plus, and COCO retrieval.

- **Jina-CLIP-V1** (Koukounas et al., 2024a) is a unified contrastive language–image model that also serves as a strong text retriever: using EVA02 ViT-B/16 as the image encoder and JinaBERT v2 as the text encoder in a staged training pipeline, it jointly optimizes image–text and text–text objectives.

- **Jina-CLIP-V2** (Koukounas et al., 2024b) is a multilingual dual-encoder vision–language model (XLM-RoBERTa text tower + EVA02-L/14 vision tower; 865M params) trained with multi-task contrastive objectives over text–text, image–text, and hard-negative triplets. It employs Matryoshka representations for flexible embedding sizes and higher-resolution training for document images, yielding strong retrieval performance in English and across 30 languages (including ViDoRe), while remaining openly available for reproducible research.

- **NEV** (Nussbaum et al., 2024) is an open weights image embedding model that shares a unified latent space with nomic embed text via a LiT style recipe that freezes the text encoder while adapting an EVA02 ViT B/16 vision tower. Trained on a large curated web corpus for multiple epochs, it targets strong zero shot classification and cross modal retrieval, reporting gains over CLIP baselines across ImageNet, DataComp, and MTEB style evaluations and providing a practical unified embedding space for vision, language, and multimodal tasks.

- **E5-V** (Jiang et al., 2024) is a multimodal embedding model that maps images, text, and interleaved inputs into a single semantic space using a prompt-based representation (for example, summarizing content in one word), which bridges the modality gap without multimodal fine-tuning. Trained only on text pairs with a contrastive objective while removing the visual pathway during training for major efficiency gains, it transfers at inference to image and mixed modality inputs and delivers strong zero-shot results on text and image retrieval, composed image retrieval, image to image retrieval with rendered text, and standard sentence similarity benchmarks.

- **MM-Embed** (Lin et al., 2025) is a universal multimodal retriever built on MLLMs that unifies text, images, and interleaved inputs; it introduces modality-aware hard negative mining and continuous fine-tuning to curb MLLM modality bias and bolster text retrieval, achieving state-of-the-art results on M-BEIR and surpassing NV-Embed-v1 on MTEB.

- **Ola** (Liu et al., 2025) is an omnimodal language model for unified image, video, and audio understanding that uses native resolution visual encoding with a Local Global Attention Pooling layer for efficient token reduction, integrates a dual audio encoder with Whisper v3 for speech and BEATs for music along with simple MLP connectors to project all modalities into a shared token space, and emphasizes cross modal alignment by treating video as the central bridge within a progressive training schedule to balance modalities.

- **Qwen2-VL** (Wang et al., 2024a) is a family of open-weight vision–language models (2B/8B/72B) that replaces fixed-resolution pipelines with Naive Dynamic Resolution and

fuses multimodal positions via M-RoPE, achieving state-of-the-art perception across images and long videos and results comparable to GPT-4o and Claude 3.5 on key benchmarks.

- **InternVL-3** (Zhu et al., 2025) is an open-source multimodal large language model that natively unifies vision and language via a single pretraining stage, avoiding post-hoc adapters and alignment. Built on a ViT–MLP–LLM stack with Variable Visual Position Encoding for long-context perception, it delivers state-of-the-art open-source results across diverse multimodal benchmarks.

- **Gemini-2.5-Flash** (AI, 2024) is a multimodal, low-latency model optimized for fast, cost-efficient inference across text, code, vision, and audio. It supports streaming generation, tool use, and extended context, making it a strong choice for interactive agents and high-throughput production systems where responsiveness is prioritized over peak accuracy.

- **GPT5** (OpenAI, 2025) is a next-generation generative pre-trained transformer that advances reliability, reasoning, and multimodal understanding. It integrates longer-context modeling with robust tool use (e.g., function calling and retrieval) and a safety-focused post-training pipeline to improve calibration and control. Together, these capabilities make GPT-5 a practical foundation for research and applications requiring dependable, grounded generation.

## D.4 EXPERIMENTAL CODE

To promote transparency and ensure the reproducibility of our work, we will release all experimental code, datasets, and detailed tutorials necessary for replicating our experiments. Our goal is to make it straightforward for researchers and practitioners to reproduce our results, regardless of their technical background. Additionally, by providing comprehensive documentation and clear guidelines, we aim to facilitate the extension of our method to other models and architectures, enabling the broader research community to explore its potential applications and improvements. We believe that open and reproducible research is essential for advancing the field and fostering collaboration.

## E CONTEXT-WINDOW LIMITATION ANALYSIS

A natural alternative to retrieval is to directly encode all items into the long context of frontier MLLMs (e.g., GPT, Gemini) and then perform end-to-end reasoning. However, this strategy is computationally prohibitive due to the quadratic growth of input tokens across heterogeneous modalities.

**Tokenization cost.** Based on Gemini's official tokenization rules, the total token budget is

$$T_{\text{total}} = 258 \sum_{m=1}^{M} \left\lceil \frac{w_m}{768} \right\rceil \left\lceil \frac{h_m}{768} \right\rceil + 263 \sum_{n=1}^{N} L_n + T_{\text{text}},$$

where $(w_m, h_m)$ are image dimensions in pixels, $L_n$ is the duration of the $n$-th video in seconds, and $T_{\text{text}}$ is the number of textual tokens. Each $768 \times 768$ image patch costs approximately 258 tokens, while each second of video costs about 263 tokens.

Our benchmark contains 46,260 multimodal items (images, videos, and documents). Even under conservative assumptions—rescaling images to a single patch and compressing videos to low frame rates—the total budget reaches nearly 200M tokens. This exceeds the largest publicly available context window (1M tokens) by more than two orders of magnitude. In practice, many items are larger than a single patch or longer than a few seconds, which pushes the requirement even higher.

This analysis highlights a fundamental limitation: even with million-token context windows, brute-force ingestion cannot approximate real-world conditions. Without targeted evidence selection, the input size scales linearly with corpus size but quadratically with attention, making end-to-end encoding infeasible. Therefore, MultiHaystack plays a critical role by providing a realistic evaluation setting where retrieval, rather than ever-larger context windows, is the decisive factor for scalable multimodal reasoning.

# F    MORE ANALYSIS

## F.1    ZERO-CONTEXT VISUAL QUESTION ANSWERING

One might suggest evaluating a zero-context baseline ($k = 0$), where the model answers questions without any retrieved content. However, this setting is fundamentally incompatible with MultiHaystack. During construction, we explicitly apply a retrieval-independence filter to remove any question that could be answered from prior knowledge or common sense alone. As shown in Table F.1, we evaluate several models under the zero-context setting, and find that even the strongest one (GPT-5) achieves only 4.28% accuracy, confirming that virtually all queries require retrieval to be solvable. Consequently, reporting $k = 0$ is not meaningful and would only obscure the purpose of the benchmark,

Table F.1: **VQA performance in zero context.**

| Model | Overall |
|---|---|
| Ola | 0.54 |
| InternVL-3 | 0.80 |
| Qwen2-VL | 0.67 |
| Gemini-2.5-Flash | 1.07 |
| GPT-5 | 4.28 |

which is to disentangle retrieval and reasoning in multimodal contexts. Instead, we provide *Gold in Top-1/5* results (Table 6), where the ground-truth item is guaranteed to be retrieved. These serve as a principled upper bound, directly isolating reasoning ability under perfect retrieval, and thus provide a far more informative diagnostic than an artificial zero-context baseline.

## F.2    PARAMETER SELECTION FOR DATA ENRICHMENT.

A central challenge in data enrichment is to retain informative positives while suppressing noisy distractors. To this end, we first apply a coarse CLIP threshold (e.g., 0.2) to discard obviously unrelated candidates. We then compute the mean CLIP similarity for positive pairs ($\approx 0.74$) and select a principled interval around it. In our dataset, this corresponds to [0.64, 0.84], which is broad enough to preserve the majority of true positives while excluding distractors with artificially high similarity or positives with abnormally low similarity. Figure F.1 highlights this separation: the purple curve

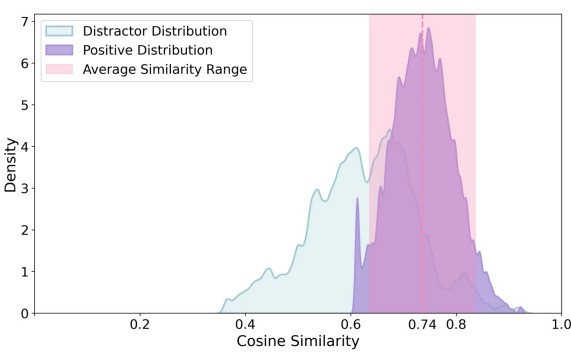

Figure F.1: **Distribution of cosine similarity scores.**

shows CLIP-based similarities between each question and its ground-truth positive image, peaking near 0.74, while the green curve shows `vidore/colqwen2-v0.1`-based similarities between each question and a large pool of candidate distractors, concentrated at lower values. By explicitly grounding the threshold in the empirical distributions of CLIP positives and `vidore/colqwen2-v0.1` distractors, this procedure yields a cleaner candidate pool, mitigating CLIP-only bias and reducing noise propagation, ultimately stabilizing downstream training.

## F.3    ADVANCED MODELS

We further evaluated the newer models InternVL-3.5 and Qwen2.5-VL (see Table F.2), and both remain well below their single-modality upper bounds once retrieval is involved, confirming that even the latest models show the same limitations.

Table F.2: **VQA performance.** Each model answers questions using top-5 items retrieved from cross-modality inputs; gray numbers show single-modality Recall@5 for reference.

| Model | Video | Image | Document | Overall |
|---|---|---|---|---|
| InternVL-3.5 | 20.95 (26.67) | 30.95 (39.72) | 51.20 (54.55) | 35.21 (42.03) |
| Qwen2.5-VL | 19.05 (21.90) | 22.17 (28.41) | 31.10 (34.45) | 24.23 (29.18) |

# G    ERROR ANALYSIS

To gain deeper insights into the failure of the current VLMs and MLLMs, we further perform a qualitative error analysis. We first compute the statistical distribution of the two major error categories: *retrieval errors* and *reasoning errors*. As shown in Figure G.1, retrieval errors account for a larger proportion overall, reflecting the difficulty of grounding queries in subtle but decisive evidence. Reasoning errors, though fewer, remain substantial, highlighting that even with correct retrieval,

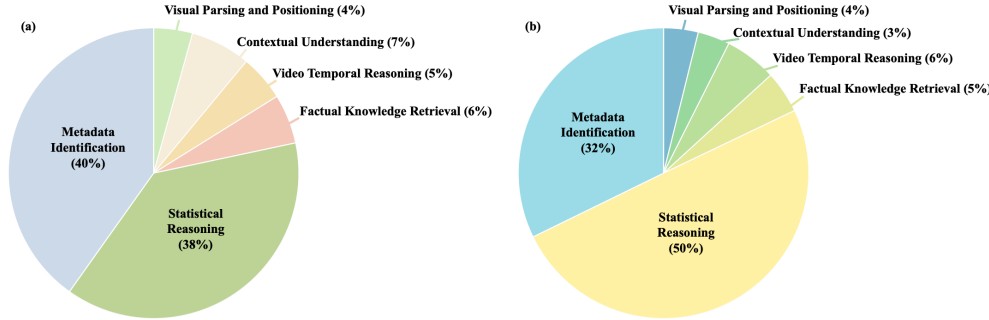

Figure G.1: **Distribution of error types.** Panels: (a) retrieval error distribution and (b) reasoning error distribution. Retrieval errors are quantified by Recall@5 with the strongest retriever (E5-V), while reasoning errors are evaluated using VQA with the strongest reasoning model (GPT-5). Retrieval errors dominate across tasks, though reasoning errors remain substantial.

models frequently fail to extract or align fine-grained content. This distribution underscores that progress in both retrieval and reasoning is necessary to reduce failure rates.

Building on this distributional view, we next examine representative cases across six tasks: Contextual Understanding, Factual Knowledge Retrieval, Metadata Identification, Statistical Reasoning, Video Temporal Reasoning, and Visual Parsing and Positioning. Figure G.2–G.7 illustrate typical examples. Retrieval errors commonly arise when models are biased toward salient but irrelevant signals (e.g., league logos, headlines, colorful infographics), overlooking subtle yet decisive cues such as timestamps or spatial relations. Reasoning errors, on the other hand, often stem from shallow associative processing, where the system outputs plausible but incorrect answers (e.g., predicting "State Farm" instead of the correct sponsor, misreporting 2,743 instead of 2,740, or confusing the spatial relation between characters). These examples reveal a consistent bottleneck: current models struggle with sensitivity to fine-grained task-relevant details, both at the retrieval and reasoning stages.

## G.1 CONTEXTUAL UNDERSTANDING (CU)

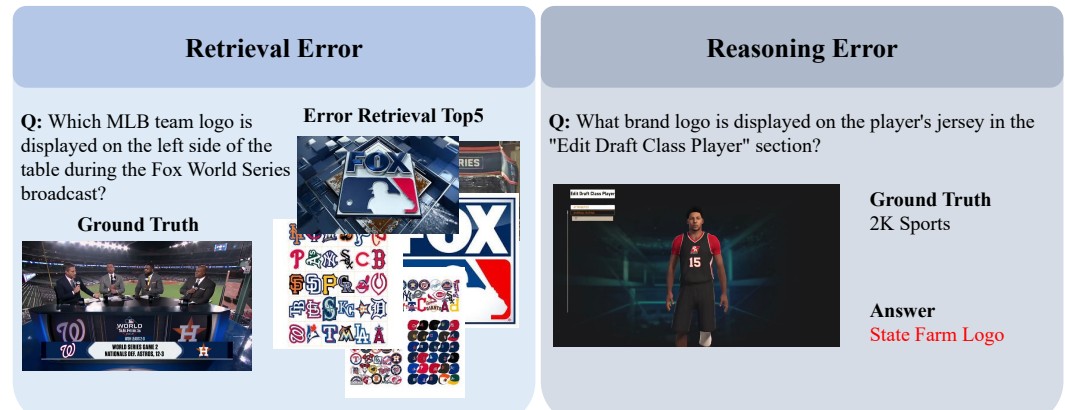

Figure G.2: **Contextual Understanding representative error cases.**

**Retrieval error.** Contextual understanding requires models to attend to subtle textual or symbolic signals embedded in a scene. As shown in Figure G.2, the retriever frequently selects broadcast frames with prominent Fox or MLB league logos, while failing to prioritize the smaller team emblem on the desk that is key to answering the query. This reveals a systematic bias toward globally salient elements and insufficient sensitivity to localized cues that define context.

**Reasoning error.** Even when the relevant frame is retrieved, models often fail to identify the intended target. In the jersey example, the system outputs "State Farm", a frequent sponsor in sports scenes, instead of the actual "2K Sports" logo. This demonstrates shallow associative reasoning, where models rely on prior familiarity with common patterns rather than aligning their predictions with the fine-grained evidence present in the scene.

## G.2 FACTUAL KNOWLEDGE RETRIEVAL (FKR)

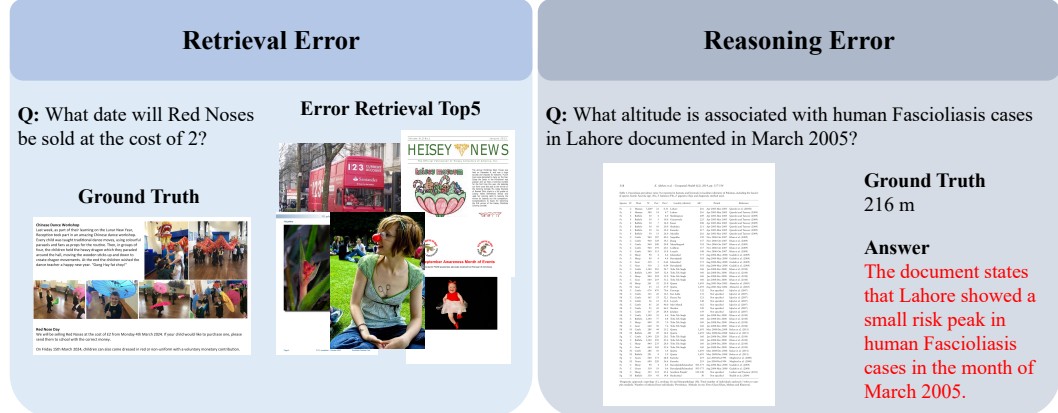

Figure G.3: **Factual Knowledge Retrieval representative error cases.**

**Retrieval error.** Factual knowledge retrieval tasks demand grounding in specific factual sources rather than surface similarity. Figure G.3 shows that retrievers often select generic news articles with overlapping topics, while missing the ownership chart that directly encodes the required fact. This indicates difficulty in filtering out visually or lexically similar distractors that lack factual relevance.

**Reasoning error.** When the correct evidence is retrieved, models may still produce factually incorrect outputs. In the card-game case, the system outputs "You can't attack this turn" instead of the precise rule "Lose 2 life." Such errors reflect limited capacity to extract exact symbolic content when distractors are semantically close or when plausible but incorrect alternatives exist in the model's training distribution.

## G.3 METADATA IDENTIFICATION (MI)

Figure G.4: **Metadata Identification representative error cases.**

**Retrieval error.** Metadata identification tasks emphasize peripheral information such as dates, publishers, or attribution details. As shown in Figure G.4, the retriever often selects documents with salient but irrelevant headlines (e.g., "Heisey News"), while failing to identify the document that actually contains the event date. This suggests that subtle metadata cues are systematically underweighted during retrieval.

**Reasoning error.** Even with the correct source, models may paraphrase broader contextual information instead of pinpointing the requested metadata. In the example, the system discusses risk levels but fails to extract the precise altitude value of 216 m. This highlights the difficulty in focusing on small but decisive details, especially when they appear in dense or noisy layouts.

## G.4 STATISTICAL REASONING (SR)

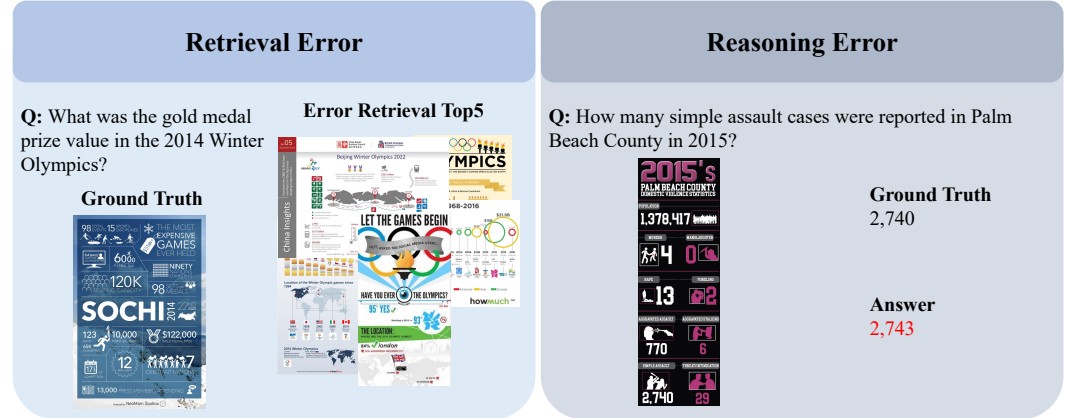

Figure G.5: **Statistical Reasoning representative error cases.**

**Retrieval error.** Statistical reasoning tasks hinge on retrieving charts or tables with exact quantitative relevance. Figure G.5 shows that retrievers sometimes surface colorful but semantically irrelevant infographics, prioritizing layout or style over the numerical semantics that matter for the query. This reveals a gap in embedding models' ability to encode quantitative intent.

**Reasoning error.** Once the correct chart is retrieved, errors often stem from fragile visual numeracy. The system may miscount bars, misalign values with axes, or confuse close numbers (e.g., reporting 2,743 instead of 2,740). Such mistakes indicate that while models perceive the chart, their mapping from visual encodings to precise numerical answers is brittle and error-prone.

## G.5 VIDEO TEMPORAL REASONING (VTR)

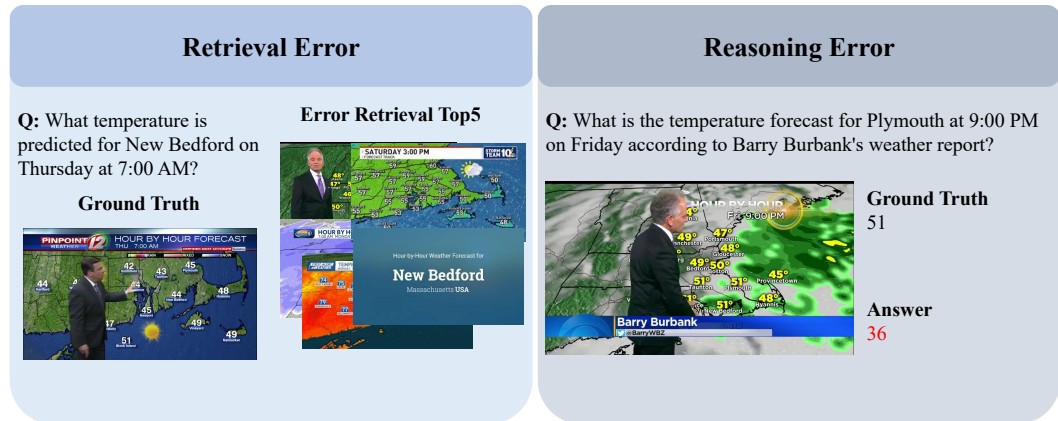

Figure G.6: **Video Temporal Reasoning representative error cases.**

**Retrieval error.** Video temporal reasoning tasks require isolating evidence at the correct temporal point. As illustrated in Figure G.6, the retriever often selects weather maps with similar layouts but corresponding to the wrong time or location, failing to encode temporal anchors. This points to the underrepresentation of sequential and time-sensitive features in retrieval embeddings.

**Reasoning error.** Even when the correct video frame is retrieved, the model may misread numeric overlays or confuse temporal ordering, e.g., predicting "36" instead of "51." These errors demonstrate the fragility of temporal–numerical reasoning, where minor OCR-like mistakes or misinterpretations of frame order propagate into incorrect conclusions.

## G.6 VISUAL PARSING AND POSITIONING (VPP)

Figure G.7: **Visual Parsing and Positioning representative error cases.**

**Retrieval error.** Visual parsing and positioning requires attention to spatial relationships rather than global scene similarity. Figure G.7 shows retrieval returning indoor scenes with similar textures or objects (e.g., laundry baskets, storage rooms) instead of the specific bag-on-table instance. This reflects insufficient encoding of spatial layout information in the retrieval stage.

**Reasoning error.** When the relevant scene is retrieved, reasoning errors arise from misinterpreting spatial relations. The model identifies the wrong character ("Skater") instead of "Baymax" when asked about the figure to the right of Stitch, showing that relational parsing across entities remains a bottleneck even when object recognition is accurate.

## G.7 SUMMARY OF ERROR PATTERNS

Across the six tasks, two consistent tendencies emerge. Retrieval errors are predominantly driven by saliency bias: systems privilege visually prominent elements such as logos, headlines, or colorful charts while neglecting the subtle but decisive cues that ground context, such as timestamps, metadata, or spatial layouts. This suggests that current multimodal embeddings fail to adequately encode task-specific contextual signals that are less obvious but more critical.

Reasoning errors, in contrast, often reflect shallow associative processing. Models default to frequent or plausible outputs, common sponsors in sports broadcasts, approximate numbers in charts, or generic spatial relations, instead of extracting the exact information encoded in the evidence. These patterns indicate that while retrieval and reasoning failures manifest differently, both are rooted in insufficient sensitivity to fine-grained, task-relevant details that determine correctness. Addressing this limitation will require embedding models that better capture subtle contextual cues and reasoning modules that enforce tighter alignment between queries and retrieved evidence.

# H ARTIFACTS AND LICENSES

We report a list of licenses for all datasets and models used in our experiment in Table H.1. We strictly follow all the model licenses and limit the scope of these models to academic research only.

**Practical usability.** In addition to licensing, we emphasize several practical aspects of dataset usability. First, the benchmark involves over 40K multimodal files (videos, images, and documents), which requires significant storage (on the order of terabytes) and compute resources for full-scale evaluation. Second, while all datasets are publicly hosted on Hugging Face under open licenses (Apache, MIT, CC BY), certain redistribution restrictions (e.g., CC-BY-NC) limit commercial use. Third, video corpora may present bandwidth challenges, and we recommend that academic users to selectively download subsets for targeted experiments. Finally, to ensure long-term accessibility, we will maintain mirrors for all datasets and scripts, together with versioned releases to facilitate

Table H.1: License information for the scientific artifacts.

| Data Sources | URL | License |
|---|---|---|
| VideoVista | Link | Apache-2.0 |
| MMBench-Video | Link | CC BY 4.0 |
| FineVideo | Link | CC BY 4.0 |
| MVBench | Link | MIT |
| DocHaystack | Link | MIT |
| MMIU | Link | CC BY 4.0 |
| A-OKVQA | Link | Apache-2.0 |
| MINT1T | Link | CC BY 4.0 |
| **Software Code / Models** | **URL** | **License** |
| CLIP | Link | MIT |
| SigLIP2 | Link | Apache-2.0 |
| OpenCLIP | Link | MIT |
| Jina-CLIP-V1 | Link | Apache-2.0 |
| Jina-CLIP-V2 | Link | CC-BY-NC-4.0 |
| NEV | Link | Apache-2.0 |
| E5-V | Link | Apache-2.0 |
| MM-Embed | Link | CC-BY-NC-4.0 |
| Ola | Link | Apache-2.0 |
| Qwen2-VL | Link | Apache-2.0 |
| InternVL-3 | Link | Apache-2.0 |
| Gemini-2.5-Flash | Link | Google Terms of Use |
| GPT-5/4o-mini | Link | OpenAI Terms of Use |

reproducibility. These considerations ensure that our benchmark is both legally compliant and practically usable by the research community.

# I    LIMITATIONS AND FUTURE WORK

Our study has several limitations. First, while MultiHaystack integrates text, images, and videos, it does not yet cover modalities such as audio or sensor signals. Extending to these would increase realism but also introduce challenges like temporal alignment and redundancy modeling. Second, benchmark construction relies on semi-automatic question generation and human verification. Although we enforce unique ground truths, annotation noise or bias may remain. Moreover, while GPT-4o is the backbone for both data construction and evaluation, we mitigate potential bias through multi-stage filtering, human checks, and consistency validation against human judgments, substantially reducing dependence on a single model. Future work could explore more scalable and diverse verification pipelines. Finally, current results are bounded by retriever quality: poor recall limits downstream reasoning regardless of model ability. Exploring retrieval-augmented training, adaptive candidate selection, or hybrid retrieval strategies may help overcome this bottleneck.

# J    BROADER IMPACT

By providing a large-scale multimodal benchmark, MultiHaystack can accelerate research on retrieval-augmented reasoning, enabling applications in search, education, healthcare, and scientific discovery. Improved systems may broaden access to complex multimodal information and support more reliable decision-making. At the same time, stronger retrieval and reasoning also raise risks, such as exposing sensitive information or amplifying misinformation. While our benchmark itself does not contain harmful content, responsible use of models evaluated on it requires privacy safeguards, robust verification, and appropriate policy frameworks. We hope MultiHaystack will guide both technical progress and responsible discourse on the societal impact of multimodal AI.

## K LLM USAGE STATEMENT

During dataset construction, we leveraged large language models as an auxiliary tool to suggest candidate question–answer pairs and to aid preliminary filtering. These outputs were then subjected to rigorous multi-stage manual verification to ensure both accuracy and diversity. For evaluation, we employed an automatic judging protocol, where an LLM was used to assist in assessing the correctness of answers from multiple VQA models. To validate the robustness of this approach, we performed direct comparisons against independent human annotations and confirmed high consistency. Finally, the manuscript underwent multiple rounds of refinement, combining careful manual revision with selective automated editing support to further improve clarity, coherence, and readability.

