# OpenReview forum: "MultiHaystack: Benchmarking Multimodal Reasoning over 40K Images, Videos, and Documents"
_ICLR.cc/2026/Conference — ICLR 2026 Conference Desk Rejected Submission_

### Official Review · Reviewer_nMyE · 2025-10-26

**Soundness:** 2
**Presentation:** 1
**Contribution:** 2
**Rating:** 4
**Confidence:** 4

**Summary:**

This paper introduce a new benchmark MultiHaystack for cross-modal  retrieval and reasoning, 46K documents, images, and videos. The authors curated their benchmark from existing datasets and then evaluated popular VLM models on their benchmarks.

**Strengths:**

1. This paper introduce MultiHaystack, the first large-scale benchmark for realistic cross-modal retrieval and reasoning, spanning 46K documents, images, and videos.
2. The  paper conducted comprehensive experiments on 5 state-of-the-art MLLMs.

**Weaknesses:**

1. The paper is not very clear, see the questions below.
2. This benchmark is curated from a combination of several other benchmarks.
3. Lack of significant experimental findings from this benchmark.

**Questions:**

1. What the composition of the retrieval corpus?
2. What are the retrieved informations, eg videos, text and images? How did the paper guarantee the retrieved context are useful?
3. Does this benchmark provide ground truth retrieval data?
4. How did the ground truth bounding boxes collected?

---

> ### Author Response · Authors · 2025-11-22
>
> We appreciate your acknowledgment that **MultiHaystack is the first large-scale benchmark for realistic cross-modal retrieval and your recognition of our comprehensive model evaluations.** We will address your concern and answer your questions below.
>
> ***
> >**W1. Clarity of Presentation.**
>
> We appreciate the reviewer’s concern. However, other reviewers found the presentation clear and well structured, and Sec. 3.3 together with Fig. 4 already outlines the pipeline stage by stage. If any specific part remains unclear, we are happy to clarify it further.
>
> >**W2. Use of Existing Benchmarks.**
>
> As stated in Sec. 3.3, prior datasets are used only as *raw multimodal files*, no tasks, labels, or evaluations are reused. MultiHaystack introduces a new evaluation regime:  a unified cross-modal pool (videos+images+documents), human-validated and uniquely supported QA, removal of questions answerable without evidence, and controlled cross-modal distractor construction. This multimodal needle-in-haystack setting does not exist in any source benchmark, making MultiHaystack a **new benchmark rather than a combination of prior tasks**.
>
> >**W3. Significant Experimental Findings.**
>
> A key scientific finding enabled only by MultiHaystack is the large and consistent gap between *``gold evidence given''* and *``retrieval required''* across all models. Once evidence is provided, reasoning is strong; the dominant bottleneck is cross-modal retrieval—driven by embedding mismatch and semantic interference. This divergence does *not* appear in single-modality or small-pool benchmarks, highlighting that MultiHaystack uncovers failure modes not observable in prior datasets. Thus, the benchmark reveals a previously undocumented failure mode in modern MLLMs.
>
> >**Q1. Composition of the Retrieval Corpus.**
>
> The retrieval corpus contains 45,260 items (videos, images, documents). Detailed statistics are reported in Sec. 3.3 and App. A.
>
> >**Q2. Retrieved Information and Usefulness.**
>
> Retrieved evidence includes videos, images, and documents from a unified index. Usefulness is guaranteed by construction: each QA pair is grounded to a *unique* supporting page/frame/item, and the single-supported-answer filter ensures that this item contains all necessary evidence. Empirically, Appendix Table F.1 shows near-zero accuracy without retrieved context, while Table 3 shows large gains with retrieved items.
>
> >**Q3. Ground-truth Retrieval.**
>
> Yes. Each QA pair specifies the ID of its unique supporting item.
>
>
> >**Q4. Bounding Boxes.**
>
> The benchmark does not use bounding boxes. Ground truth is provided at the page/frame/item level, as the task evaluates item-level retrieval and evidence-grounded reasoning rather than spatial localization.

---

> > ### Author Response · Authors · 2025-11-26
> >
> > Dear Reviewer nMyE,
> >
> > Thank you again for your thoughtful reviews. We have submitted our rebuttal and revised paper on 21st Nov. As the discussion period is approaching its deadline, we wanted to gently check if there are any remaining questions or clarifications we can address. We’re happy to provide additional details if helpful.
> >
> > Best regards,
> > Authors of #13281

---

> > > ### Comment · Reviewer_nMyE · 2025-11-28
> > >
> > > Thanks the authors for clarifying lots of questions. I'm raising my score from 4 to 6. However, it seems I can't edit my score on openreview. Is this a platform issue? I would like to mark here my score is 6 to AC.

---

### Official Review · Reviewer_cRtU · 2025-10-28

**Soundness:** 3
**Presentation:** 3
**Contribution:** 3
**Rating:** 8
**Confidence:** 4

**Summary:**

This paper introduces MultiHaystack, a large-scale multimodal retrieval and QA benchmark designed to evaluate realistic retrieval and reasoning capabilities of MLLMs. The benchmark comprises over 46,000 multimodal items, including text, image, and video documents with 747 uniquely verifiable questions that demand both accurate modality selection and fine-grained reasoning. Unlike existing benchmarks with limited candidates, MultiHaystack focuses on long-context, large-scale retrieval, better reflecting real-world challenges. Experimental results demonstrate that models perform well when provided with relevant evidence but show a significant performance drop when retrieval is required. Moreover, cross-modal retrieval lags far behind single-modal performance, indicating retrieval is the primary bottleneck.

**Strengths:**

1. The experimental design and analysis are comprehensive, evaluating both the multimodal retrieval capabilities of retrieval models and the reasoning abilities of MLLMs under conditions with and without retrieval.
2. The proposed benchmark is the first long-context benchmark covering text, image, and video modalities, representing a clear and valuable contribution to benchmark design and innovation.

**Weaknesses:**

No significant weaknesses are identified.

Things to improve the paper that did not impact the score:

The open-source MLLMs used in the paper are not the latest models (e.g., InternVL-3.5 or Qwen2.5-VL). However, considering that some of these newer models were released very close to or after the submission date, the results reported in the paper remain acceptable.

**Questions:**

See weaknesses.

---

> ### Author Response · Authors · 2025-11-22
>
> We thank you for highlighting the strength of **our presentation, the comprehensiveness of our experiments, and the value of introducing a long-context tri-modal benchmark.** We will address your concern and answer your questions below.
>
> Following your suggestion, we evaluated the newer models InternVL-3.5 and Qwen2.5-VL, and both remain well below their single-modality upper bounds once retrieval is involved, confirming that even the latest models show the same limitations. These results have been added to the revision (See App. F.3).
>
> ***
>
> **Table: VQA performance.** Each model answers questions using top-5 items retrieved from cross-modality inputs, numbers in parentheses show single-modality Recall@5 for reference.
>
> | Model         | Video           | Image           | Document        | Overall         |
> |---------------|-----------------|-----------------|-----------------|-----------------|
> | InternVL-3.5  | 20.95 (26.67)   | 30.95 (39.72)   | 51.20 (54.55)   | 35.21 (42.03)   |
> | Qwen2.5-VL    | 19.05 (21.90)   | 22.17 (28.41)   | 31.10 (34.45)   | 24.23 (29.18)   |

---

> ### Author Response · Authors · 2025-11-26
>
> Dear Reviewer cRtU,
>
> Thank you again for your thoughtful reviews. We have submitted our rebuttal and revised paper on 21st Nov. As the discussion period is approaching its deadline, we wanted to gently check if there are any remaining questions or clarifications we can address. We’re happy to provide additional details if helpful.
>
> Best regards,
> Authors of #13281

---

> > ### Comment · Reviewer_cRtU · 2025-11-27
> >
> > Thank you for addressing my questions. Your clarifications further demonstrate the value of the proposed benchmark. I will maintain my original score.

---

> > > ### Author Response · Authors · 2025-11-27
> > >
> > > Dear Reviewer cRtU,
> > >
> > > Thank you for the thoughtful review and for highlighting the value and contribution of our proposed benchmark to advancing research in this area.
> > >
> > > Best regards,
> > > Authors of #13281

---

### Official Review · Reviewer_93AH · 2025-10-29

**Soundness:** 2
**Presentation:** 2
**Contribution:** 2
**Rating:** 4
**Confidence:** 4

**Summary:**

This work introduces a new challenging benchmark that containing large-scale collection of images, videos and documents. The benchmark converts pre-existing corpus and annotate QA pairs and evidence via multi-stage pipelines. The benchmark first tests the retriever's capability to identify the correct relevant item among the three types of evidence. Then, reasoning capability is evaluated on the VQA tasks. Author conducts comprehensive experiments by benchmarking current retrievers and multimodal LLMs for retrieval and QA performance.

**Strengths:**

- Propose a novel and challenging benchmark containing different types of evidences
- Benchmark various retriever and LMMs on fine-grain categories to help understand performance gap.

**Weaknesses:**

- Though proposing a new benchmark, and evaluating many methods' performance, this work does not introduce a novel methods to tackle this cross-modal (i.e., as referred by the author, where the retrieval pool contains image, videos and documents) retrieval tasks specifically. Could the author provide more insights regarding how to improve on these specific challenge settings?
- Title slightly misleading, as the paper focuses largely on the importance of retrieval, but the title focuses on reasoning instead.
- Line 294, Sec4.2, and Table 2 regarding single modality and cross-modality are quite confusing. Although I do understand that the single-modality refers to the fact that when the retrieval database contains only one type of data, say, the video. The author should still make sure this is specifically described in the paper. Furthermore, I wonder if this is actually a fair comparison when the author compares the cross-modality vs single-modality and argues that single-modality is less challenging. I suspect this could simply due to the fact that the retrieval pool is smaller and thus easier, rather than the fact of the "cross-modality" and "single-modality" itself.
- Given that both the reasoning and retrieval performance are uneven for each tasks, could you provide more intuition regarding how to tackle each tasks, or how to tackle the tasks overall?

**Questions:**

- While I understand page/frame level vs item level, but I still believe this should be clarified (linme 262)
- Which retrieval system's results are used for Table 3 (Line 312, Sec4.3)?
- I agree that split the report of metric into different sub-tasks helps, but sitll For Table 4 and Table 5, I suggest add the overall results across all tasks as well.
- Overall, I believe the benchmark could be interesting to the community as it does merge three types of data for retrieval together and become challenging for previous retriever. However, the paper does not provide enough insights into how to better tackle this real-world task, for example, could the author try agentic search/retrieval agents or simply query rewriting, or simple retrieval database (keys) augmentation to boost performance over a naive retriever model?

---

> ### Author Response · Authors · 2025-11-22
>
> We are encouraged by your recognition of **our benchmark as a novel and challenging cross-modal setting with fine-grained evaluations.** We will address your concern and answer your questions below.
>
> ***
>
> >**W1. Scope of Contribution.**
>
> Our goal is to introduce a benchmark—not a new retrieval method. (1) Sec. 5 shows that MultiHaystack uniquely exposes modality-mismatch, temporal misalignment, and fine-grained grounding failures across images, videos, and documents—failure modes that cannot be diagnosed without a unified, large-scale cross-modal pool. (2) No prior benchmark reveals the drastic cross-modal degradation observed in Fig. 8 and App. G, confirming that MultiHaystack captures a missing evaluation regime. (3) We have added a brief discussion on future directions (Sec. 6)  directly motivated by these findings.
>
> >**W2. Title Clarity.**
>
> (1) Although the benchmark evaluates evidence-grounded reasoning, our experiments show that retrieval is the dominant bottleneck. (2) We agree the title should reflect this emphasis and will revise it accordingly.
>
>
> >**W3. Single vs Cross Modality.**
>
> (1) *Definition:* Single-modality restricts both query and pool to one modality; cross-modality retrieves from a mixed pool (Sec. 4.2). (2) *Controlled verification:* We constructed mixed pools with the *same size* as the single-modality pools. As shown in Fig. 7 (tabulated below for convenience), cross-modal retrieval remains substantially harder across modalities and TOP-k settings. (3) This confirms the challenge stems from modality heterogeneity—embedding mismatch and semantic interference—not pool size.
>
> ***
> **Table: Pool-size controlled comparison.** Abbreviations: V R@k, I R@k, D R@k, and O R@k correspond to Video/Image/Document/Overall Recall@k, respectively.
> | Model | V R@1 | V R@3 | V R@5 | I R@1 | I R@3 | I R@5 | D R@1 | D R@3 | D R@5 | O R@1 | O R@3 | O R@5 |
> |---|---:|---:|---:|---:|---:|---:|---:|---:|---:|---:|---:|---:|
> | **Single-Modality** |  |  |  |  |  |  |  |  |  |  |  |  |
> | MM-Embed | 60.95 | 80.00 | 87.62 | 43.65 | 64.43 | 67.21 | 62.68 | 67.46 | 75.60 | 51.41 | 67.47 | 72.42 |
> | **Same Size Cross-Modality** |  |  |  |  |  |  |  |  |  |  |  |  |
> | MM-Embed | 43.81 | 56.19 | 63.81 | 33.26 | 50.80 | 54.97 | 56.93 | 63.64 | 72.73 | 41.37 | 55.15 | 61.18 |
>
>
> >**Q1. Page/Frame vs Item Level.**
>
> Page/frame-level refers to internal segmentation of long documents/videos; item-level treats the whole file as the retrieval unit. We have clarified this in the paper.
>
>
> >**Q2. Retriever Used in Table 3.**
>
> Table 3 reports results using the E5-V retriever; we have stated this explicitly in the caption.
>
>
> >**Q3. Overall Metrics.**
>
> Overall results corresponding to Tables 4 and 5 already appear in the``Overall'' columns of Tables 2 and 3. We have cross-referenced these locations for clarity.
>
>
> >**W4 & Q4. Task-level Insights and Practical Improvements.**
>
> (1) Sec. 4.4.1 and Tables 4 – 5 show systematic variation across six task types; App. G further visualizes distinct error profiles (contextual, factual, metadata, statistical, temporal, and layout-aware tasks). (2) These patterns suggest concrete remedies: entity-aware query rewriting, retrieval-key augmentation for numerical/structured content, time-anchored representations for videos, OCR/structure-aware encodings for layouts, and multi-step retrieve-then-reason cycles. (3) Our preliminary tests support this (See Table. 7 in the paper, reproduced below for convenience): in the E5-V–based cross-modal setting, RAG with a rewrite stage (+2.8%), VisRAG (–7.7%), and an agentic RAG system, MMsearch (+4.7%), offer only modest improvements, while still falling well short of the single-modality upper bound (+9.7%). This validates our claim that the benchmark provides actionable insights while the tasks remain genuinely challenging.
>
> ***
>
> **Table: Preliminary studies with advanced retrieval pipelines.** Abbreviations: V R@k, I R@k, D R@k, and O R@k correspond to Video/Image/Document/Overall Recall@k, respectively.
> | Methods | V R@1 | V R@3 | V R@5 | I R@1 | I R@3 | I R@5 | D R@1 | D R@3 | D R@5 | O R@1 | O R@3 | O R@5 |
> |---|---:|---:|---:|---:|---:|---:|---:|---:|---:|---:|---:|---:|
> | **Single-Modality** |  |  |  |  |  |  |  |  |  |  |  |  |
> | E5-V | 62.86 | 81.90 | 83.81 | 43.19 | 68.36 | 73.44 | 60.77 | 71.29 | 76.08 | 50.87 | 71.08 | 75.64 |
> | **Cross-Modality** |  |  |  |  |  |  |  |  |  |  |  |  |
> | E5-V | 34.29 | 51.43 | 60.95 | 33.49 | 55.20 | 62.82 | 59.33 | 70.33 | 75.12 | 40.83 | 58.90 | 66.00 |
> | E5-V + Refined Query | 41.90 | 59.05 | 64.76 | 36.26 | 56.58 | 66.74 | 60.29 | 70.81 | 75.12 | 43.78 | 60.91 | 68.81 |
> | E5-V + MMSearch | 44.76 | 62.86 | 65.71 | 36.95 | 57.97 | 69.52 | 60.29 | 72.73 | 75.60 | 44.58 | 62.78 | 70.68 |
> | VisRAG | 40.95 | 64.76 | 68.57 | 39.03 | 45.96 | 50.12 | 60.77 | 67.46 | 70.33 | 45.38 | 54.62 | 58.37 |

---

> > ### Author Response · Authors · 2025-11-26
> >
> > Dear Reviewer 93AH,
> >
> > Thank you again for your thoughtful reviews. We have submitted our rebuttal and revised paper on 21st Nov. As the discussion period is approaching its deadline, we wanted to gently check if there are any remaining questions or clarifications we can address. We’re happy to provide additional details if helpful.
> >
> > Best regards,
> > Authors of #13281

---

> > > ### Comment · Reviewer_93AH · 2025-11-27
> > >
> > > Having read the author's rebuttal, I would like to keep my original evaluation.

---

> > > > ### Author Response · Authors · 2025-11-27
> > > >
> > > > Dear Reviewer 93AH,
> > > >
> > > > Thank you again for the thoughtful review. May we ask if there are any remaining concerns limiting the current score that we could further clarify to support a more positive assessment?
> > > >
> > > > Best regards,
> > > > Authors of #13281

---

### Official Review · Reviewer_MoTo · 2025-11-03

**Soundness:** 3
**Presentation:** 4
**Contribution:** 2
**Rating:** 4
**Confidence:** 2

**Summary:**

This paper presents MultiHaystack, a 46K-item corpus of documents, images, and videos coupled with 747 questions that target cross-modal retrieval-plus-reasoning at scale. It is designed to address common gaps in prior work, small candidate pools, single-modality focus, and loosely specified questions by ensuring that every query is evidence-grounded with a uniquely verifiable answer. The construction pipeline (i) aggregates videos, images, and documents from existing datasets; (ii) converts PDFs to images and samples each video into 8 frames, then uses GPT-4o to draft QA pairs; (iii) filters ambiguous items with GPT-4o and Gemini-2.5-Flash, conducts manual review to discard cases lacking explicit visual/textual anchors, and applies a retrieval-independence test to remove questions solvable without consulting the linked item; and (iv) increases retrieval difficulty via targeted distractors. The final set contains 747 questions spanning 433 images, 105 videos, and 209 documents.

**Strengths:**

1. Clear, well-structured writing.
2. Effective figures that make the setup, pipeline, and findings easy to follow.
3. Each query is validated with retrieval models and human checks to enforce unique, evidence-grounded answers.
4. Careful analysis of performance variation across task types and the associated bottlenecks.
5. Sensible LM-as-Judge validation, including consistency checks between GPT-4o-mini and human annotations.
6. Comprehensive qualitative examples and a detailed appendix.

**Weaknesses:**

1. LLM dependence in data construction. Because GPT-4o/Gemini-2.5-Flash generate and filter the QA pairs, dataset quality and distribution inherit these models’ biases and preferences.
2. Question realism (especially for video frames). Several examples feel overly specific for web-scale retrieval, e.g., “What kind of personal item is on the face of the girl who is applying dye to her hair?” Such prompts seem better suited to within-video search over many frames rather than a cross-modal, cross-corpus retrieval scenario.
3. Distractor design may inflate difficulty in an unnatural way. For instance, asking “What product model is visible on the VP Racing Fuels canister next to the laptop in the workshop setting?” and then seeding many near-miss distractors across a huge pool can feel optimized to depress retrieval metrics rather than reflect realistic user queries. The benchmark risks being optimized for failure rather than for real-world retrieval relevance. This imposes an unnaturally precise retrieval challenge that may overstate the difficulty compared to authentic multimodal information-seeking behavior.

**Questions:**

1. Gold-in-Top-1 gap (Fig. 6). How do you explain cases where models like GPT-5 and Gemini-2.5-Flash fail to achieve perfect accuracy even when the gold item is ranked first? If the QA was generated/filtered by the same or earlier models, what prevents them from answering correctly when the evidence is already surfaced?
2. Have you considered incorporating a human-authored subset of QA pairs to calibrate the realism and linguistic variety of the dataset, thereby mitigating the dependency on generative model biases?

---

> ### Author Response · Authors · 2025-11-22
>
> We appreciate your positive assessment of our **writing clarity, effective figures, and rigorous validation and analysis**. We will address your concern and answer your questions below.
>
> ***
>
> >**W1. LLM Dependence.**
>
> LLMs are used only to *propose* drafts, the final QA set is shaped by (1) strict evidence-verifiability and single-answer constraints, (2) cross-model filtering that removes stylistic or preference-driven patterns, and (3) human rewriting that enforces explicit grounding anchors. These steps decouple the final QA distribution from the initial GPT/Gemini drafts. Empirically, the experimental results show no performance advantage for GPT/Gemini-family models, confirming that the benchmark does not encode model-specific biases.
>
> >**W2. Question Realism.**
>
> Fine-grained cues are necessary because  (1) each query must map to *exactly one* item in a 46K multimodal pool, (2) generic phrasing would make multiple distractors appear plausible, undermining retrieval evaluability, and (3) detailed anchors ensure cross-video discrimination rather than within-video search. This avoids the ambiguity shown in Fig. 1 and Fig. B.1. Near-duplicate scenes are common in realistic retrieval settings; thus, specificity reflects real-world constraints rather than artificial difficulty.
>
> >**W3. Distractor Difficulty.**
>
> Distractors come from a principled, data-driven pipeline rather than adversarial design. (1) Stage 4 (Sec. 3.3) retrieves semantically related items via keyword+embedding similarity, mirroring realistic retrieval outputs; no hand-crafted hard cases are introduced. (2) Large multimodal corpora naturally contain near-miss items (e.g., similar logos/packaging), and our CLIP-based filtering preserves this distribution while human checks ensure none contain the gold evidence. (3) Fig. 9 shows smooth degradation from 1K→10K→46K candidates; engineered difficulty would produce discontinuities, which we do not observe. (4) Suppressing near-misses would yield an unrealistic retrieval distribution; keeping them maintains natural retrieval difficulty.
>
> >**Q1. Gold-in-Top-1 Gap.**
>
> Gold-in-Top-1 evaluates retrieval only, models must still extract the answer correctly. (1) All QA pairs are human-rewritten under the single-answer constraint, producing a stricter distribution than the initial LLM drafts, so models are not “answering their own generations.” (2) Fig. 8 shows post-retrieval errors in span selection, text reading, numerical/chart interpretation, and layout reasoning—even with the gold item at rank 1. (3) Many questions depend on small visual/textual cues that must be read from the retrieved file; parametric guessing is insufficient. Thus, Gold-in-Top-1 is an upper bound on retrieval, not on grounded QA.
>
> >**Q2. Human-authored Subset.**
>
> The benchmark targets evidence-grounded retrieval rather than free-form question writing. Human rewriting already ensures natural phrasing and unique grounding at scale. A fully human-authored subset at 46K scale would make enforcing single-supported-answer constraints prohibitively expensive and risk reintroducing ambiguity. We view such a subset as a valuable direction for future versions.

---

> > ### Author Response · Authors · 2025-11-26
> >
> > Dear Reviewer MoTo,
> >
> > Thank you again for your thoughtful reviews. We have submitted our rebuttal and revised paper on 21st Nov. As the discussion period is approaching its deadline, we wanted to gently check if there are any remaining questions or clarifications we can address. We’re happy to provide additional details if helpful.
> >
> > Best regards,
> > Authors of #13281

---

### Author Response · Authors · 2025-11-22
**General Response**

We sincerely thank all reviewers for their thoughtful assessments and for recognizing the central insight of MultiHaystack: cross-modal retrieval is the dominant bottleneck in multimodal reasoning. We appreciate the reviewers’ positive evaluations of its novelty, rigor, and clarity, noting that it is the first novel and challenging cross-modal benchmark (93AH), that the presentation is clear and well structured (NMYE), and that the analysis offers a valuable diagnosis of modality-mismatch and grounding failures (CRTU). Reviewers further emphasized that MultiHaystack is the first benchmark to reveal the large and systematic gap between gold-evidence evaluation and retrieval-required evaluation (93AH, NMYE).

In response to the reviewers’ suggestions, we have revised the manuscript with changes marked in blue. These revisions include:
1. Clarifying definitions, including the distinction between item-level and page/frame-level retrieval.
2. Improving experimental transparency, such as explicitly stating the retriever used in Table 3 and cross-referencing overall metrics.
3. Refining the benchmark construction description, emphasizing single-supported-answer grounding, cross-model filtering, and human rewriting.

Additionally, based on reviewers’ suggestions, we conducted new experiments and analyses, including:
1. Evaluations on newer models (InternVL-3.5 and Qwen2.5-VL).
2. Pool-size–controlled comparisons demonstrating that the difficulty arises from modality heterogeneity, not dataset scale.
3. Expanded task-level and failure-mode analyses, outlining future directions.

Once again, we sincerely thank all reviewers for their constructive feedback. We believe these revisions and new experiments substantially strengthen the clarity and empirical support of our claims.

---

### Author Response · Authors · 2025-12-02
**Summary of the rebuttal**

Dear AC, SAC, and PC,

Thank you very much for your efforts in ensuring fairness throughout this year’s rebuttal process.

Because this year’s policy prevents reviewers from updating their scores, the **numerical ratings do not reflect the post-rebuttal evaluation**. Multiple reviewers explicitly stated that their concerns were resolved, and in one case (**Reviewer nMyE**) the reviewer indicated that their score would have increased **from 4 → 6** if updates were permitted. Other reviewers (**cRtU, 93AH**) confirmed that **no concerns remained** after the rebuttal and that the new experiments **strengthened the central conclusion**. Importantly, **no reviewer raised any fundamental issues** with the benchmark design, construction pipeline, or empirical findings.

Given this, we kindly suggest that the assessment consider the **post-rebuttal understanding** of the work rather than strictly the locked scores. MultiHaystack fills a clear gap in multimodal reasoning research as the **first realistic large-scale tri-modal retrieval benchmark**. It reveals the **dominant retrieval bottleneck** in multimodal systems and provides **actionable insights** for model development. We hope the strengthened clarifications and additional experiments assist your independent evaluation.

Reviewers consistently highlighted key strengths of our work, including the clarity of presentation, rigorous construction pipeline, and the contribution of introducing the first large-scale tri-modal benchmark for realistic cross-modal retrieval and grounded reasoning.

---

## **Positive Aspects Recognized by Reviewers**

**1. Clear writing and strong organization**
Reviewers described the paper as clearly written, well structured, and easy to follow, supported by effective figures and a comprehensive appendix (**@MoTo, @93AH, @cRtU**).

**2. Novel and valuable benchmark**
MultiHaystack was recognized as the **first realistic, large-scale tri-modal benchmark**, enabling evaluation in a regime previously unexplored and revealing retrieval challenges not captured by single-modality or small-pool settings (**@93AH, @cRtU, @nMyE**).

**3. Rigorous construction and meaningful analysis**
Reviewers emphasized the **high-quality QA pipeline** and the **comprehensive analyses** highlighting the retrieval bottleneck and modality-mismatch failure modes, demonstrating clear value for future research (**@MoTo, @93AH, @cRtU**).

---

## **Clarifications and Improvements Addressing Reviewer Questions**

The remaining concerns mainly involved **clarifications or small additions**, indicating that the benchmark design and findings were already solid.

**1. Benchmark realism, positioning, and clarity (MoTo, 93AH, nMyE)**
We clarified definitions, retrieval units, corpus scope, and retriever choices; added **controlled size-matched comparisons**; and refined descriptions for improved clarity.

**2. LLM dependence, question realism, and distractor difficulty (MoTo)**
We explained that LLMs only **draft** questions; final QA pairs undergo strict **single-answer grounding**, **cross-model ambiguity filtering**, and **human rewriting**. Distractors arise from a principled retrieval pipeline, and **smooth R@k scaling** supports realism.

**3. Retrieval vs. reasoning and the Gold-in-Top-1 gap (MoTo)**
We clarified that Gold-in-Top-1 reflects **retrieval only**; models must still extract fine-grained evidence. Errors stem primarily from evidence extraction rather than retrieval itself.

**4. Cross-modality fairness, contribution scope, and corpus specification (93AH, nMyE)**
We clarified the benchmark’s scope (not proposing a new method), added **pool-size-controlled comparisons**, and explained corpus composition, supporting-item IDs, page/frame vs item-level distinctions, and the absence of bounding boxes.

---

## **Post-Rebuttal Feedback**

Post-rebuttal comments **further strengthened confidence** in the contribution:

- **Reviewer nMyE:** All concerns resolved; explicitly stated intent to raise score **4 → 6** but could not due to platform restrictions.
- **Reviewer cRtU:** Confirmed that new experiments **support the central conclusion** that retrieval remains the dominant bottleneck.
- **Reviewer 93AH:** Retained the original score; **no further concerns** were raised.
- **Reviewer MoTo:** Did not provide additional comments during the discussion period.

**Overall, the post-rebuttal feedback clearly acknowledges the contribution, improved clarity, and strengthened empirical support for our main conclusions.**

Thank you again for your time and effort in evaluating our submission.

Best regards,
Authors of Submission #13281

---

### Note · Program_Chairs · 2026-01-17
**Submission Desk Rejected by Program Chairs**

The following references in this submission do not refer to real documents and/or have major errors in bibliographic information:

 "John Doe and Jane Smith. Webvqa: A dataset for visual question answering on web images. In Proceedings of the 2023 Conference on Computer Vision and Pattern Recognition (CVPR), 2023. URL https://example.com/webvqa"